# Chloride-dependent mechanisms of multimodal sensory discrimination and nociceptive sensitization in *Drosophila*

Nathaniel J Himmel, Akira Sakurai, Atit A Patel, Shatabdi Bhattacharjee, Jamin M Letcher, Maggie N Benson, Thomas R Gray, Gennady S Cymbalyuk, Daniel N Cox*

Neuroscience Institute, Georgia State University, Atlanta, Georgia

**Abstract** Individual sensory neurons can be tuned to many stimuli, each driving unique, stimulus-relevant behaviors, and the ability of multimodal nociceptor neurons to discriminate between potentially harmful and innocuous stimuli is broadly important for organismal survival. Moreover, disruptions in the capacity to differentiate between noxious and innocuous stimuli can result in neuropathic pain. *Drosophila* larval class III (CIII) neurons are peripheral noxious cold nociceptors and innocuous touch mechanosensors; high levels of activation drive cold-evoked contraction (CT) behavior, while low levels of activation result in a suite of touch-associated behaviors. However, it is unknown what molecular factors underlie CIII multimodality. Here, we show that the TMEM16/anoctamins *subdued* and *white walker* (*wwk; CG15270*) are required for cold-evoked CT, but not for touch-associated behavior, indicating a conserved role for anoctamins in nociception. We also evidence that CIII neurons make use of atypical depolarizing chloride currents to encode cold, and that overexpression of *ncc69*—a fly homologue of *NKCC1*—results in phenotypes consistent with neuropathic sensitization, including behavioral sensitization and neuronal hyperexcitability, making *Drosophila* CIII neurons a candidate system for future studies of the basic mechanisms underlying neuropathic pain.

*For correspondence:
dcox18@gsu.edu

Competing interest: The authors declare that no competing interests exist.

## Editor's evaluation

This is an important manuscript that clarifies mechanisms of multimodality in a class of insect somatosensory neurons and presents a model for how Cl⁻ currents underlie cold nociception. The authors support the claims in the paper through the use of gene knockdown, behavioral experiments, neuroanatomy, and optogenetic activation in the *Drosophila* fruit fly. The demonstration that the same class of somatosensory neurons can respond to innocuous versus noxious stimuli, depending upon which protein those neurons are activated, could shed light on disease states such as neuropathic pain when touch signals are confused as painful.

## Introduction

Noxious stimuli are transduced by high-threshold sensory neurons referred to as nociceptors, and these sensory neurons often respond to more than one stimulus type—a property called sensory poly—or multimodality (*Himmel et al., 2017*; *Le Bars et al., 2001*; *Sherrington, 1906*; *Smith and Lewin, 2009*; *Sneddon, 2004*). Although the ability to differentiate between sensory modalities is inarguably important for organismal survival, it remains relatively poorly understood what molecular mechanisms facilitate sensory multimodality within single neurons or neural subtypes; and these systems are of direct importance for human health, as the inability to discriminate between noxious

and innocuous stimuli is thought to underlie chronic neuropathic pain (*Basbaum et al., 2009*; *Lumpkin and Caterina, 2007*; *Woolf and Ma, 2007*).

Like other animals, *Drosophila melanogaster* senses and responds to noxious stimuli. In larvae, nociception begins in peripheral dendritic arborization neurons of the class III (CIII) and class IV (CIV) subtypes. CIV neurons are polymodal high-temperature, mechanical, and chemical (menthol and acid) nociceptors, and activation of CIV neurons by any one of these sensory modalities elicits a corkscrew-like rolling behavior (*Himmel et al., 2019*; *Hwang et al., 2007*; *Lopez-Bellido et al., 2019*; *Neely et al., 2011*; *Tracey et al., 2003*). In contrast, CIII neurons are multimodal cold nociceptors (*Himmel et al., 2021*; *Maksymchuk et al., 2022*) and innocuous touch mechanosensors (*Tsubouchi et al., 2012*; *Yan et al., 2013*). Activation of CIII neurons drives stimulus-specific behaviors: acute noxious cold primarily elicits a highly stereotyped, bilateral contraction (CT) response, wherein the body rapidly shortens along the head-to-tail axis (*Turner et al., 2016*), while innocuous touches elicit a suite of behaviors, including head-withdrawal, head casting/turning behavior, and reverse locomotion (*Kernan et al., 1994*).

CIII neurons function via a high-low threshold detection system, whereby high levels of CIII activation (and strong $Ca^{2+}$ transients) are associated with CT, and low levels of activation (and relatively modest $Ca^{2+}$ transients) with touch behaviors. This is evidenced in $Ca^{2+}$ imaging experiments and via optogenetics, where strong optogenetic activation of CIII neurons elicits CT, and less strong activation primarily drives head withdrawals (*Turner et al., 2016*). However, molecular factors underlying this high-low threshold filter have not been elucidated. Given the relatively constrained function and modality-specific behavioral outputs of CIII, this system constitutes a good target for elucidating mechanisms underlying multimodality in single neural classes.

Stimulus-evoked CIII calcium transients are thought to occur as a result of the activation of transient receptor potential (TRP) channels, a superfamily of variably selective cation channels which participate in nociception across a wide variety of species (*Himmel and Cox, 2020*). In *Drosophila*, the TRP genes, *Pkd2*, *NompC*, and *Trpm,* are required for cold nociception and innocuous touch mechanosensation (*Turner et al., 2016*). In vertebrates, $Ca^{2+}$-dependent channels of a variety of subtypes (e.g. anoctamin/TMEM16 channels; *Takayama et al., 2015*) interact with TRP channels in nociceptive systems in order to drive appropriate levels of neural activation, making them a potential candidate mechanism underlying cold and touch discrimination. Previous studies have shown that the *Drosophila* gene *subdued* encodes an anoctamin/TMEM16 $Ca^{2+}$-activated $Cl^-$ channel (CaCC) (*Le et al., 2019*; *Wong et al., 2013*), and that it is necessary for high-temperature-evoked rolling (*Jang et al., 2015*). However, it is unknown whether subdued might function to encode stimulus-specific sensory information in multimodal neurons. We therefore questioned whether anoctamins might function in CIII in a modality-specific fashion.

In the present study, we tested the hypothesis that anoctamins expressed in CIII are required for cold nociception. We have found that the anoctamins *subdued* and *white walker* (*wwk; CG15270*)—here shown to be orthologous to human ANO1/2 and ANO8, respectively—are required for cold nociception. Interestingly, anoctamins participate in an excitatory capacity, suggesting that CIII neurons make use of excitatory $Cl^-$ currents, and we provide additional evidence that CIII neurons likely use depolarizing $Cl^-$ currents to selectively encode cold. Furthermore, we demonstrate that overexpression of *ncc69* (a fly homologue of *NKCC1*) is sufficient for driving sensitization of cold nociception in *Drosophila* larvae.

## Results

### The anoctamins *subdued* and *CG15270* are expressed in *Drosophila* CIII cold nociceptors

Previous work has demonstrated that TRP channels are required for both CIII-dependent cold nociception and innocuous touch mechanosensation (*Turner et al., 2016*); however, no genes have been identified, which are selectively required for CIII cold nociception. As vertebrate TRP channels function alongside anoctamin/TMEM16 channels in vertebrate nociceptors (*Takayama et al., 2015*), we first sought to identify anoctamins which might selectively participate in *Drosophila* cold nociception.

Cell-type-specific transcriptomic data indicate that the anoctamin genes *subdued* and *CG15270* (*Figure 1A*) are enriched in CIII cold nociceptors, as compared to average expression in whole larvae

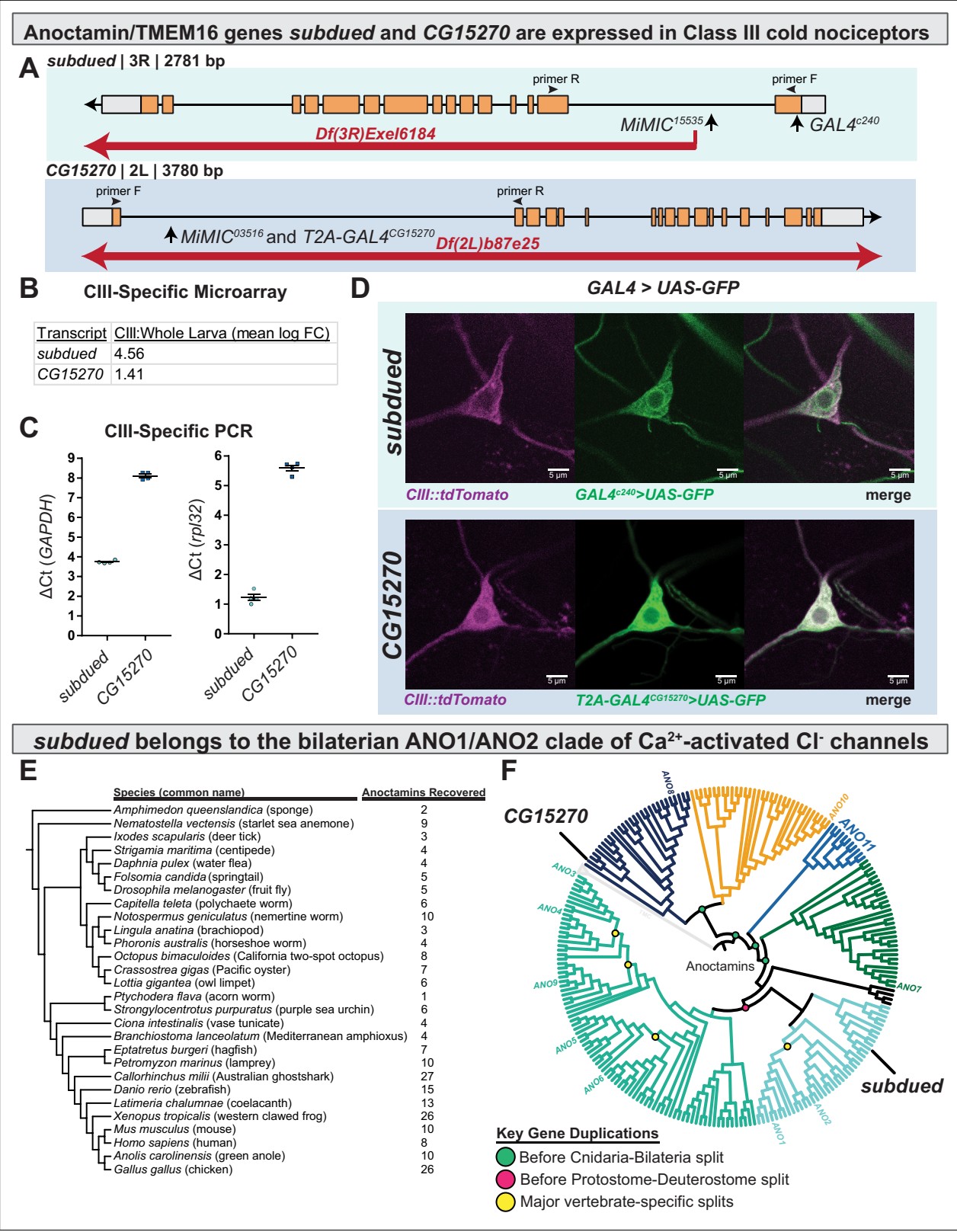

**Anoctamin/TMEM16 genes *subdued* and *CG15270* are expressed in Class III cold nociceptors**

**A** *subdued* | 3R | 2781 bp

primer R
primer F

*Df(3R)Exel6184*

*MiMIC*[15535]

*GAL4*[c240]

*CG15270* | 2L | 3780 bp

primer F
primer R

*MiMIC*[03516] and *T2A-GAL4*[CG15270] *Df(2L)b87e25*

**B** CIII-Specific Microarray

| Transcript | CIII:Whole Larva (mean log FC) |
|---|---|
| *subdued* | 4.56 |
| *CG15270* | 1.41 |

**C** CIII-Specific PCR

ΔCt (*GAPDH*)   ΔCt (*rpl32*)

*subdued*   *CG15270*

**D** *GAL4 > UAS-GFP*

*subdued*

CIII::tdTomato   *GAL4*[c240]*>UAS-GFP*   merge

*CG15270*

CIII::tdTomato   *T2A-GAL4*[CG15270]*>UAS-GFP*   merge

**subdued belongs to the bilaterian ANO1/ANO2 clade of Ca²⁺-activated Cl⁻ channels**

**E**

| Species (common name) | Anoctamins Recovered |
|---|---|
| *Amphimedon queenslandica* (sponge) | 2 |
| *Nematostella vectensis* (starlet sea anemone) | 9 |
| *Ixodes scapularis* (deer tick) | 3 |
| *Strigamia maritima* (centipede) | 4 |
| *Daphnia pulex* (water flea) | 4 |
| *Folsomia candida* (springtail) | 5 |
| *Drosophila melanogaster* (fruit fly) | 5 |
| *Capitella teleta* (polychaete worm) | 6 |
| *Notospermus geniculatus* (nemertine worm) | 10 |
| *Lingula anatina* (brachiopod) | 3 |
| *Phoronis australis* (horseshoe worm) | 4 |
| *Octopus bimaculoides* (California two-spot octopus) | 8 |
| *Crassostrea gigas* (Pacific oyster) | 7 |
| *Lottia gigantea* (owl limpet) | 6 |
| *Ptychodera flava* (acorn worm) | 1 |
| *Strongylocentrotus purpuratus* (purple sea urchin) | 6 |
| *Ciona intestinalis* (vase tunicate) | 4 |
| *Branchiostoma lanceolatum* (Mediterranean amphioxus) | 4 |
| *Eptatretus burgeri* (hagfish) | 7 |
| *Petromyzon marinus* (lamprey) | 10 |
| *Callorhinchus milii* (Australian ghostshark) | 27 |
| *Danio rerio* (zebrafish) | 15 |
| *Latimeria chalumnae* (coelacanth) | 13 |
| *Xenopus tropicalis* (western clawed frog) | 26 |
| *Mus musculus* (mouse) | 10 |
| *Homo sapiens* (human) | 8 |
| *Anolis carolinensis* (green anole) | 10 |
| *Gallus gallus* (chicken) | 26 |

**F**

*CG15270*

ANO8   ANO10   ANO11

ANO3

ANO4

Anoctamins

ANO7

ANO9

ANO5

ANO6   ANO1   ANO2

*subdued*

Key Gene Duplications
● Before Cnidaria-Bilateria split
● Before Protostome-Deuterostome split
● Major vertebrate-specific splits

**Figure 1.** *subdued* and *white walker* (*CG15270*) are anoctamin/TMEM16 channels expressed in class III (CIII) neurons. (**A**) Alleles, chromosomal deficiencies, and enhancer trap or T2A *GAL4s* used in this study, as well as approximate location for primers (**F and R**) used to validate mutants. (**B**) Class III expression of *subdued* and *CG15270* from cell-type-specific microarray (GSE69353), expressed as mean log fold-change difference between isolated CIII and whole-larval samples. Positive values indicate enrichment/upregulation and negative values downregulation. (**C**) qRT-PCR validating CIII

*Figure 1 continued on next page*

*Figure 1 continued*

expression of *subdued* and *CG15270* (n=4, each condition). (**D**) *UAS-GFP* driven under the control of a *subdued* enhancer trap, *GAL4^{c240}* or *T2A-GAL4^{wwk}*, evidences that *subdued* and *white walker* are expressed in CIII neurons. (**E**) Species, cladogram, and number of sequences used in phylogenetic analysis. (**F**) Maximum likelihood phylogeny of animal anoctamins, with weak branches rearranged in a species-aware manner against cladogram in E. *subdued* is a member of the ANO1/2 clade of nephrozoan calcium-activated chloride channels and of the clade of metazoan anoctamins that includes mammalian ANO1/2/3/4/5/6/9. Mammalian ANO8 is homologous to *CG15270*. Additionally, this phylogeny evidences a separate clade of ANOs of unknown function we have deemed ANO11 (cyan).

The online version of this article includes the following figure supplement(s) for figure 1:

**Figure supplement 1.** Overlapping expression of *GAL4^{nompC}* and the newly described class III (*CIII::tdTomato*) fusion.

**Figure supplement 2.** *GAL4^{c240}* driven *UAS-GFP* expression.

**Figure supplement 3.** *T2A-GAL4^{wwk}* driven *UAS-GFP* expression.

**Figure supplement 4.** Maximum likelihood phylogeny of anoctamins corresponding to *Figure 1*.

**Figure supplement 5.** NOTUNG-generated phylogeny of anoctamins corresponding to *Figure 1*.

(*Figure 1B*; GEO accession GSE69353). To independently validate CIII expression of these genes, we performed qRT-PCR on isolated CIII neurons and found detectable expression of *subdued* and *CG15270* consistent with their relative abundance as revealed by the transcriptomic data (*Figure 1C*). To further validate these data, we drove expression of green fluoroescent protein (GFP) using enhancer trap or T2A *GAL4*s for *subdued* and *CG15270*, respectively, pairing these with a *CIII::tdTomato* fusion line to confirm CIII expression (*Figure 1—figure supplement 1*). For *subdued*, we used the previously described *GAL4^{c240}*, which did in fact drive GFP expression in CIII nociceptors (*Figure 1D*). Consistent with previous reports (*Jang et al., 2015*), *GAL4^{c240}* also drove GFP expression in CIV nociceptors (*Figure 1—figure supplement 2*). For *CG15270*, we made use of a *CG15270*-specific *T2A-GAL4* (Trojan *GAL4*); this approach has been shown to be a strong indicator of gene expression (*Lee et al., 2018*). *CG15270 T2A-GAL4* drove expression of GFP in CIII nociceptors (*Figure 1D*). In fact, *CG15270* appeared to be broadly expressed in larval peripheral sensory neurons (e.g. *Figure 1—figure supplement 3A*), as well as in the larval central nervous system (*Figure 1—figure supplement 3B*).

## The evolutionary history of anoctamins supports that *subdued* is part of the ANO1/ANO2 subfamily of Ca²⁺-activated Cl⁻ channels

While previous work has phylogenetically characterized anoctamins/TMEM16 channels (*Medrano-Soto et al., 2018*; *Milenkovic et al., 2010*; *Wang et al., 2013*), their evolution and familial organization outside of vertebrates are relatively poorly understood and therefore limits our ability to formulate hypotheses of function based on phylogeny. We therefore generated an anoctamin phylogeny, which is inclusive of a wide range of animal taxa (*Figure 1E*). This phylogeny (*Figure 1F*, *Figure 1—figure supplements 4–5*) indicates that anoctamins are organized into five major metazoan subfamilies, which predate the Cnidaria-Bilateria split (including a new subfamily we deem ANO11, which has not been previously described), and six major bilaterian subfamilies. *subdued* has been previously determined to encode a CaCC (*Wong et al., 2013*); consistent with these findings, this phylogeny evidences that *subdued* is a member of the bilaterian ANO1/2 subfamily of CaCCs. *CG15270* is one of the more distantly related anoctamins and is of unknown function, but in contrast to *subdued,* has a single homologue in humans (both part of the metazoan ANO8 subfamily).

## *subdued* and *CG15270* (*white walker*) are required for cold nociception

Using the previously developed cold plate assay (*Patel and Cox, 2017*), we recorded larval behavior while delivering an acute ventral cold stimulus, which causes animals to contract (CT) (*Turner et al., 2016*) along the head-to-tail axis (*Figure 2A*). We first assessed the behavior of larvae carrying homozygous <u>Mi</u>nos-<u>m</u>ediated <u>i</u>ntegration <u>c</u>assette alleles (MiMIC, abbreviated as Mi; a transposon containing a gene-trap cassette leading to gene disruption) for *subdued* and *CG15270* (*Venken et al., 2011*; *Figure 1A*). RT-PCR analyses of MiMIC insertions for either *subdued* or *CG15270* suggest these mutations are severely hypomorphic in the expression of these genes (*Figure 2—figure supplement 1*). Homozygous mutant larvae (both *subdued* and *CG15270*) showed decreased cold sensitivity—larvae had a reduced peak magnitude of CT (measured by maximum magnitude reduction in length normalized to initial length, *Figure 2B*), and as such fewer animals strongly CT in response to noxious cold

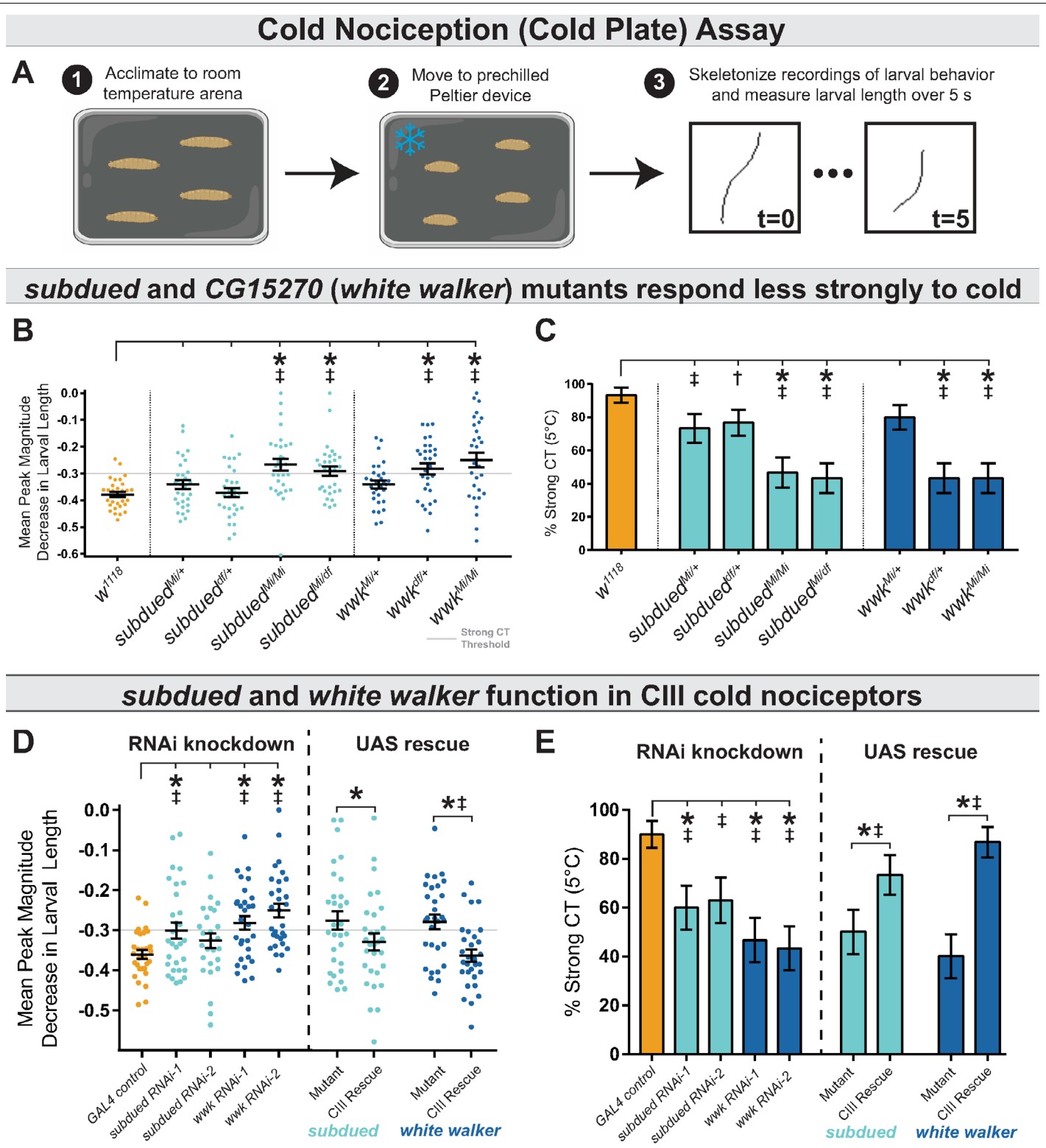

**Figure 2.** Anoctamins function in class III (CIII) neurons. (**A**) For cold plate assay, larvae were acclimated to a room temperature arena before being transferred to a pre-chilled cold plate. Contraction (CT) was identified by measuring the length of skeletonized larvae over the course of chilling. (**B**) Mutant analysis; mean peak magnitude in larval CT. $w^{1118}$ (n=30); $subdued^{Mi/+}$ (n=30; p=0.58; $BF_{10}$=1.18); $subdued^{df/+}$ (n=30; p=1; $BF_{10}$=0.28); $subdued^{Mi/Mi}$ (n=30; p<0.001; $BF_{10}$=228.63); $subdued^{Mi/df}$ (n=30; p=0.008; $BF_{10}$=737.03); $wwk^{Mi/+}$ (n=30; p=0.58; $BF_{10}$=1.60); $wwk^{df/+}$ (n=30; p=0.003; $BF_{10}$=324.11); $wwk^{Mi/Mi}$ (n=30; p<0.001; $BF_{10}$=433.18). Also see *Figures 1–3*. (**C**) Mutant analysis; % of animals which strongly CT in response to noxious

*Figure 2 continued on next page*

*Figure 2 continued*

cold (≥30% reduction in body length). Mutations in *subdued* and *white walker* result in a reduced percent of larvae which strongly CT in response to noxious cold (5°C). $w^{1118}$ (n=30); *subdued*$^{Mi/+}$ (n=30; p=0.13; $BF_{10}$=4.417); *subdued*$^{df/+}$ (n=30; p=0.25; $BF_{10}$=2.826); *subdued*$^{Mi/Mi}$ (n=30; p<0.001; $BF_{10}$=461.34); *subdued*$^{Mi/df}$ (n=30; p<0.001; $BF_{10}$=997.24); *wwk*$^{Mi/+}$ (n=30; p=0.45; $BF_{10}$=2.00); *wwk*$^{df/+}$ (n=30; p<0.001; $BF_{10}$=997.24); *wwk*$^{Mi/Mi}$ (n=30; p<0.001; $BF_{10}$=997.24). (**E**) CIII-specific knockdown and rescue analyses; mean peak magnitude in larval CT. Knockdown: GAL4 control (n=30); *subdued RNAi-1* (n=30; p=0.049; $BF_{10}$=3.98); *subdued RNAi-2* (n=27; p=0.437; $BF_{10}$=0.78); *wwk RNAi-1* (n=30; p=0.005; $BF_{10}$=89.25); *wwk RNAi-2* (n=30; p<0.001; $BF_{10}$=6932.18). Rescue: *GAL4*$^{nompC}$;*subdued*$^{Mi/Mi}$ (mutant, n=30); *GAL4*$^{nompC}$/*UAS-subdued*;*subdued*$^{Mi/Mi}$ (CIII rescue, n=30; p=0.049; $BF_{10}$=1.61). *wwk*$^{Mi/Mi}$;*GAL4*$^{19-12}$ (mutant, n=30); *wwk*$^{Mi/Mi}$;*GAL4*$^{19-12}$/*UAS-wwk* (CIII rescue, n=30; p<0.001; $BF_{10}$=83.59). (**D**) CIII-specific knockdown and rescue analyses; % of animals which strongly CT in response to noxious cold (≥30% reduction in body length). CIII-specific knockdown (*GAL4*$^{19-12}$) of *subdued* and *white walker* results in a reduced percent of larvae which strongly CT in response to noxious cold. GAL4 control (n=30); *subdued RNAi-1* (n=30; p=0.039; $BF_{10}$=7.64); *subdued RNAi-2* (n=27; p=0.075; $BF_{10}$=4.93); *wwk RNAi-1* (n=30; p=0.002; $BF_{10}$=71.17); and *wwk RNAi-2* (n=30; p<0.001; $BF_{10}$=138.15). GAL4-UAS-mediated CIII rescue of *subdued* and *white walker* in mutant backgrounds increased cold sensitivity. *GAL4*$^{nompC}$;*subdued*$^{Mi/Mi}$ (n=30); *GAL4*$^{nompC}$/*UAS-subdued*;*subdued*$^{Mi/Mi}$ (n=30; p=0.031; $BF_{10}$=3.57). *wwk*$^{Mi/Mi}$;*GAL4*$^{19-12}$ (n=30); *wwk*$^{Mi/Mi}$;*GAL4*$^{19-12}$/*UAS-wwk* (n=30; p<0.001; $BF_{10}$=281.95).

The online version of this article includes the following figure supplement(s) for figure 2:

**Figure supplement 1.** Validation of mutagenic effects of *subdued* and *white walker* MiMIC mutations.

**Figure supplement 2.** Cold nociception defects of *subdued* and *white walker* mutations.

**Figure supplement 3.** Cold nociception defects of *subdued* and *white walker* RNAi in CIII cold nociceptive neurons.

**Figure supplement 4.** Rescue analyses of cold nociceptive defects for *subdued* and *white walker*.

(*Figure 2C*, *Figure 2—figure supplement 2*). These data suggest that *CG15270* and *subdued* are required for cold nociception.

We also assessed behavior in larvae carrying deficient chromosomes (large chromosomal deletions covering our genes of interest), heterozygous MiMIC mutations, and transheterozygous combinations of the MiMIC mutation over the chromosomal deficiency. Larvae bearing a heterozygous *subdued* allele or a single chromosome with a deficiency covering the gene (*df*) displayed largely normal cold sensitivity (Bayesian analyses indicating evidence of a modest effect on % strong CT in the heterozygous condition), while the mutant allele paired with the deficiency recapitulated the relatively strong homozygous phenotype (*Figure 2B–C*, *Figure 2—figure supplement 2*). This constitutes evidence against the alternative hypothesis that the *subdued* phenotype is the result of off-target mutations. Mutants carrying a heterozygous *CG15270* allele behaved relatively normal, but a single deficiency resulted in reduced cold sensitivity (*Figure 2B–C*, *Figure 2—figure supplement 2*). Furthermore, the homozygous deficiency and the deficiency over MiMIC allele were lethal. We were thus unable to rule out off-target effects for *CG15270* by this approach.

## *subdued* and *CG15270* (*white walker*) are required in CIII neurons for normal CT behavior and cold-evoked CIII activity

As *subdued* and *CG15270* mutants had impaired cold nociception, we next sought to test the hypothesis that *subdued* and *CG15270* function in CIII neurons (and further, to rule out off-target effects). To this end, we next knocked down the expression of *subdued* and *CG15270* by *UAS*-mediated RNAi, targeting CIII neurons using the *GAL4*$^{19-12}$ driver (*Turner et al., 2016*). UP-TORR (https://www.flyrnai.org/up-torr/) was used to assess computationally predicted off-target effects for these RNAi constructs (*Hu et al., 2013*); UP-TORR indicates that these constructs have no predicted off-target effects. Consistent with the mutant phenotypes, CIII-specific knockdown of *subdued* or *CG15270*—via two independent RNAi constructs each—resulted in reduced peak magnitude of CT (*Figure 2D*) and thus reduced % CT (*Figure 2E*, *Figure 2—figure supplement 3*). Although *subdued RNAi-2* showed a statistically insignificant decrease in % strong CT, the phenotype is extremely similar to *subdued RNAi-1,* and Bayesian analyses indicate that there is substantial evidence in favor of the difference. These results provide further evidence that these genes are required for cold nociception—specifically, that they are required in CIII neurons. We therefore suggest the name *white walker* (*wwk*) for *CG15270*, as *white walker* mutant larvae are less sensitive to noxious cold. For further experimentation, we made use of the RNAi line with the strongest phenotype.

We further tested the hypothesis that *subdued* and *white walker* function in CIII neurons by assessing whether the mutant phenotypes were due to specific loss of function in CIII nociceptors. We predicted that, if the mutant phenotype was due to loss of function in CIII neurons, and if *wwk* and *subdued* functioned in cold nociception in CIII neurons, then CIII-driven *UAS-white walker* or *UAS-subdued*

expression would rescue cold sensitivity in mutant backgrounds. For these experiments, we used CIII-targeting *GAL4*s located on chromosomes other than the chromosome containing the experimental gene (for *white walker*, *GAL4¹⁹⁻¹²*; for *subdued*, *GAL4ⁿᵒᵐᵖC*). GAL4-UAS-mediated expression of the anoctamins in mutant backgrounds rescued cold sensitivity (*Figure 2D–E*, *Figure 2—figure supplement 4*).

We next tested the hypothesis that anoctamins are involved in cold-evoked neural activity by making electrophysiological recordings in live, filet larvae during chilling. Consistent with previous reports (*Himmel et al., 2021*; *Maksymchuk et al., 2022*), control CIII firing frequency had an inverse relationship with temperature—the frequency of neural firing increased as temperature decreased—and change in temperature (chilling) resulted in bursting activity, while stable cold exposure resulted in tonic firing which increased in frequency with decreased temperature (*Figure 3A and D*).

GAL4-UAS-mediated, CIII-specific knockdown of *subdued* or *white walker* resulted in decreased cold-evoked CIII neural activity, particularly in steady-state tonic firing at a stable cold temperature (*Figure 3B–C*). Relative to controls, CIII-specific knockdown of *subdued* or *white walker* resulted in a reduction in overall firing frequency (spikes/s) at both 15 and 10°C (*Figure 3D*). As *subdued* is thought to encode a CaCC, we tested $Ca^{2+}$ necessity by bath application of 1,2-bis(*o*-aminophenoxy) ethane-*N,N,N',N'*-tetraacetic acid (BAPTA-AM) (although this would affect all $Ca^{2+}$-dependent cellular components and not just CaCCs). BAPTA-AM silenced neurons at 20°C, greatly reduced cold-evoked activity at 15°C, and resulted in a shift in the latency to the peak magnitude of activity at 10°C, demonstrating the deep involvement of $Ca^{2+}$ (*Figure 3E*, *Figure 3—figure supplement 1*). While latency to peak magnitude was increased with BAPTA-AM application relative to vehicle control at 10°C, we did not observe a notable suppressive effect on overall tonic spiking, which suggests that additional $Ca^{2+}$-independent mechanisms contribute to activating CIII cold nociceptive sensory neuron firing at this noxious cold temperature. *subdued* and *white walker* are not required for mechanosensation, locomotion, or dendrite morphogenesis.

We next tested the alternative hypothesis that these observed phenotypes reflect a loss of general excitability of CIII neurons. As CIII neurons are multimodal cold nociceptors and innocuous touch mechanosensors, loss of general excitability or overall neural function would result in decreased innocuous-touch mechanosensitivity; for example, previous reports have shown that silencing CIII neurons (*Tsubouchi et al., 2012*; *Yan et al., 2013*) and loss of function of select TRP channels (*Turner et al., 2016*) result in defects in innocuous-touch mechanosensitivity. Innocuous touch assays (based on the Kernan assay; *Kernan et al., 1994*) revealed that neither *subdued* or *white walker* loss of function affects innocuous-touch mechanosensitivity (*Figure 4A–B*). These results indicate that, in CIII neurons, *subdued* and *white walker* are selectively required for cold nociception.

We next tested the alternative hypothesis that knockdown leads to generally decreased contractile ability, which could affect locomotion. Neither knockdown of *subdued* nor *white walker* resulted in locomotion phenotypes (*Figure 4C–G*).

We also tested the alternative hypothesis that decreased cold sensitivity was due to morphological defects in CIII dendritic arborization. CIII neurons labeled with GFP show no dendritic defects under *subdued* and *white walker* knockdown (*Figure 4H–J*); there were no quantitative differences in the number of dendritic branches, total dendritic length, or dendritic branch density (*Figure 4K–M*).

## CIII neurons make use of excitatory Cl⁻ currents to encode noxious cold

As *subdued* encodes an ANO1/ANO2 CaCC (*Wong et al., 2013*; *Figure 1E–F*), knockdown reduces cold-evoked behavior (*Figure 2*) and neural activity (*Figure 3A–D*), and bath application of a $Ca^{2+}$ chelator affected cold-evoked neural activity (*Figure 3E*), we hypothesized that CIII neurons make use of atypical, excitatory CaCC to encode noxious cold.

Our cold plate behavior and electrophysiology results are consistent with the hypothesis that CIII Cl⁻ currents are excitatory and selectively required for responses to cold stimulation. However, these results are not alone sufficient to disprove the alternative hypothesis that Cl⁻ currents are inhibitory but have excitatory downstream effects; for example, inhibitory currents may be required for patterning neural activity in such a way to appropriately drive behavior, as is the case in *Drosophila* CIV neurons with respect to hyperpolarizing K⁺ currents (*Onodera et al., 2017*; *Terada et al., 2016*).

To more proximately test the hypothesis that Cl⁻ currents are excitatory in CIII, we drove CIII-specific expression of Aurora, an engineered Cl⁻ channel which gates in response to blue-light illumination

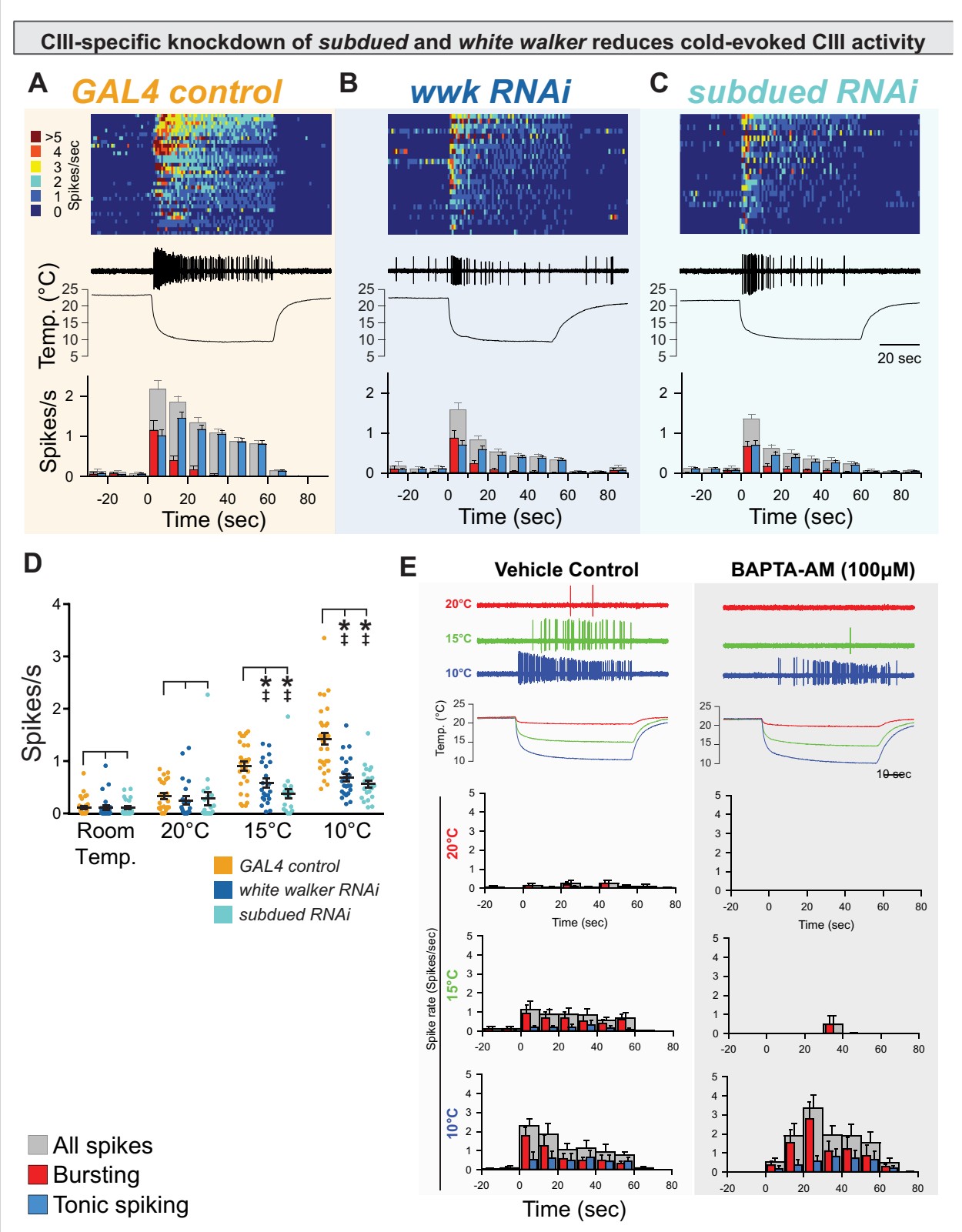

**Figure 3.** CIII-specific knockdown of *subdued* and *white walker* reduces cold-evoked neural activity. (**A, B, C**) Top: Heatmap representation of cold-evoked class III (CIII) activity (10°C), with each line representing an individual sample prep. Middle: Representative traces of cold-evoked neural activity over graph of temperature ramp. Knockdown of *subdued* and *white walker* results in decreased cold-evoked firing, particularly in the steady-state tonic firing at steady-state temperature. Bottom: Representation of average frequency from population binned by 10 s. Red and blue bars show the

*Figure 3 continued on next page*

*Figure 3 continued*

proportion of bursting vs tonic spiking activity. Knockdown of *subdued* and *white walker* results in decreased tonic firing. (**D**) CIII-specific knockdown of *subdued* and *white walker* results in decreased cold-evoked firing frequency at temperature ramps to 15 and 10°C. Room temp (N=78): GAL4 control (n=32); *white walker RNAi* (n=24; p>0.99; BF$_{10}$=0.274); *subdued RNAi* (n=22; p≥0.99; BF$_{10}$=0.279). 20°C (N=65): GAL4 control (n=25); *white walker RNAi* (n=22; p=0.69; BF$_{10}$=0.39); *subdued RNAi* (n=18; p=0.90; BF$_{10}$=0.32). 15°C (N=68): GAL4 control (n=27); *white walker RNAi* (n=22; p=0.0082; BF$_{10}$=3.64); *subdued RNAi* (n=19; p<0.001; BF$_{10}$=92.73). 10°C (N=78): GAL4 control (n=32); *white walker RNAi* (n=24; p<0.001; BF$_{10}$=3310.09); *subdued RNAi* (n=22; p<0.001; BF$_{10}$=44346.95). (**E**) Cold-evoked CIII activity in the presence of vehicle (left; n=6) or BAPTA-AM (right; n=6).

The online version of this article includes the following figure supplement(s) for figure 3:

**Figure supplement 1.** Application of BAPTA results in reduced cold sensitivity; n=6 for each condition.

(*Wietek et al., 2017*). Aurora is typically employed as an inhibitory optogenetic tool, as Cl$^-$ currents are typically hyperpolarizing/inhibitory. However, if CIII Cl$^-$ currents are excitatory, one would expect optogenetic activation of these Cl$^-$ channels to result in CIII-mediated behaviors. While blue light illumination does not normally cause contractile behavior (*Figure 5—video 1*), optogenetic activation of CIII neurons via cation-conductive ion channels (here, ChETA) results in robust CT behavior in larvae (*Figure 5*, *Figure 5—video 2*). Likewise, blue light illumination of freely locomoting *GAL4$^{19-12}$>UAS-Aurora* larvae elicited CT behavior, providing strong evidence against the alternative hypothesis that CIII Cl$^-$ currents are inhibitory (*Figure 5*, *Figure 5—video 3*). This behavior differs slightly from cation optogenetics; however, in that the Aurora-evoked CT is shorter-lived and immediately followed by touch-associated behaviors, such as head casting and reverse locomotion, perhaps suggesting that the Cl$^-$ currents alone are relatively weak.

We next asked how these currents might be excitatory. Across animal taxa, neural Cl$^-$ homeostasis is maintained by differential expression of SLC12 co-transporters (*Figure 6A*)—in *Drosophila*, *kazachoc* (*kcc*) encodes an outwardly facing K$^+$-Cl$^-$ cotransporter (*Hekmat-Scafe et al., 2006*), while *ncc69* encodes an inwardly facing Na$^+$-K$^+$-Cl$^-$ cotransporter (*Leiserson et al., 2011*). The differential expression of these cotransporters can therefore modulate the membrane-potential effects of Cl$^-$ currents. Transcriptomic data indicates that *kcc* is downregulated, and *ncc69* is upregulated in CIII neurons, as compared to average expression levels in whole larvae (*Figure 6B*; CIII expression independently validated in *Figure 6C*), suggesting that CIII neurons may maintain relatively high intracellular Cl$^-$ concentrations, thereby facilitating depolarizing (and therefore excitatory) Cl$^-$ currents. In order to manipulate Cl$^-$ homeostasis in CIII nociceptors, we used RNAi and UAS constructs to knock down and overexpress *ncc69* and *kcc* and observed effects on noxious cold-evoked behavior.

Modulating the expression of SLC12 cotransporters affected cold-evoked behavior (*Figure 6D–E*, *Figure 6—figure supplement 1*). Knockdown of *ncc69* and overexpression of *kcc*—both of which would theoretically decrease intracellular Cl$^-$ concentration—resulted in reduced % strong CT. This difference was statistically different for *ncc69 RNAi*, and there is substantial evidence according to Bayesian analyses for an effect in both *ncc69 RNAi* and *kcc OE* on % strong CT. However, support for the *kcc OE* phenotype was notably weaker, as there was no evidence for differences in average peak magnitude decrease in larval length. Furthermore, knockdown of *kcc* and overexpression of *ncc69*—which would both theoretically increase intracellular Cl$^-$ concentration—did not obviously affect sensitivity to 5°C noxious cold. Manipulating Cl$^-$ physiology in this fashion did not affect innocuous touch mechanosensitivity (*Figure 6F–G*), indicating that Cl$^-$ physiology is selectively required for CIII cold nociception.

## Low extracellular Cl$^-$ concentrations and overexpression of *ncc69* result in nociceptive sensitization

We next modulated extracellular Cl$^-$ in our electrophysiology preparations in order to more directly observe the effect of the Cl$^-$ gradient on CIII activity. Decreasing extracellular Cl$^-$ sensitized CIII neurons; bath application of low Cl$^-$ saline induced spontaneous CIII bursting and increased sensitivity to cooling (*Figure 7A*), effects which could be washed out. Relative to controls, we observed increased spontaneous firing under low Cl$^-$ saline at room temperature, increased firing at 20 and 15°C, and no effect at 10 °C (*Figure 7B*).

One possible hallmark of neuropathic pain in mammals is dysregulation of ion homeostasis (including *NKCC1*; a human orthologue of *ncc69*) in neurons involved in nociception. Such dysregulation could hypothetically result in increased intracellular Cl$^-$—and thereby aberrantly excitatory

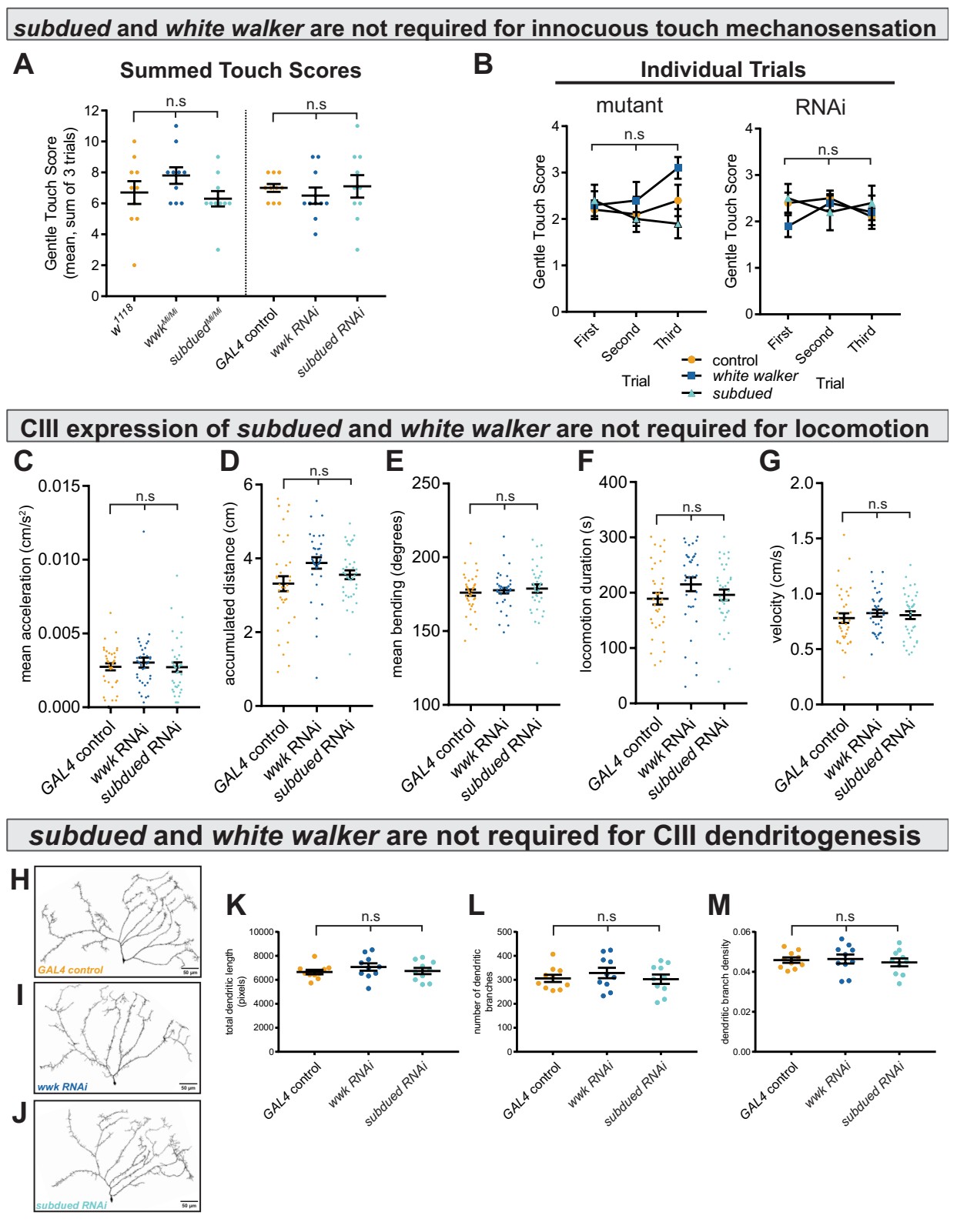

**Figure 4.** *subdued* and *white walker* are not required for innocuous mechanosensation, locomotion, or class III (CIII) dendritogenesis. (**A**) There is no difference in innocuous touch mechanosensation (sum Kernan touch scores, three trials for each sample) in either *subdued* or *white walker* mutants (n=10 for each condition; p=0.20; $BF_{10}$=0.62) or GAL4-UAS-mediated CIII-knockdown (n=10 for each condition; p=0.70; $BF_{10}$=0.27). (**B**) For each genotype, there were no within-subjects effects in innocuous touch sensation across trials. $w^{1118}$ (n=10; p=0.69; $BF_{10}$=0.27); $wwk^{Mi/Mi}$ (n=10; p=0.19;

*Figure 4 continued on next page*

*Figure 4 continued*

BF$_{10}$=0.85); *subdued$^{Mi/Mi}$* (n=10; p=0.42; BF$_{10}$=0.42); *GAL4* control (n=10; p=0.39; BF$_{10}$=0.50); *wwk RNAi* (n=10; p=0.37; BF$_{10}$=0.47); *subdued RNAi* (n=10; p=0.80; BF$_{10}$=0.25). (**C–G**) There are no differences in acceleration (p=0.59; BF$_{10}$=0.13), accumulated distance (p=0.056; BF$_{10}$=0.96), bending behavior (p=0.71; BF$_{10}$=0.025), locomotion duration (p=0.22; BF$_{10}$=0.30), or average velocity (p=0.69; BF$_{10}$=0.12), between control and knockdown (n$_{control}$ = 36, n$_{wwk}$ = 35, and n$_{subdued}$ = 36). (**H–J**) Representative neural images of CIII ddaF neuron dendritic arbors under control and knockdown conditions. (**K–M**) There is no difference in total dendritic length (p=0.48; BF$_{10}$=0.34), number of branches (p=0.70; BF$_{10}$=0.27), or dendritic branch density (p=0.81; BF$_{10}$=0.24) between control and knockdown (n=10 for each condition).

Cl$^-$ currents—leading to nociceptive sensitization and spontaneous nociceptor activity, much like we observed under low Cl$^-$ saline conditions (*Mòdol et al., 2014*; *Tan et al., 2020*). In an attempt to genetically model neuropathic pain, we overexpressed *ncc69* in CIII neurons, predicting it would likewise cause nociceptive sensitization. Overexpression of *ncc69* sensitized neurons to cooling resulted in spontaneous, room temperature CIII activity (*Figure 7C–D*). Curiously, we did not observe deficits in CIII firing under *ncc69* knockdown.

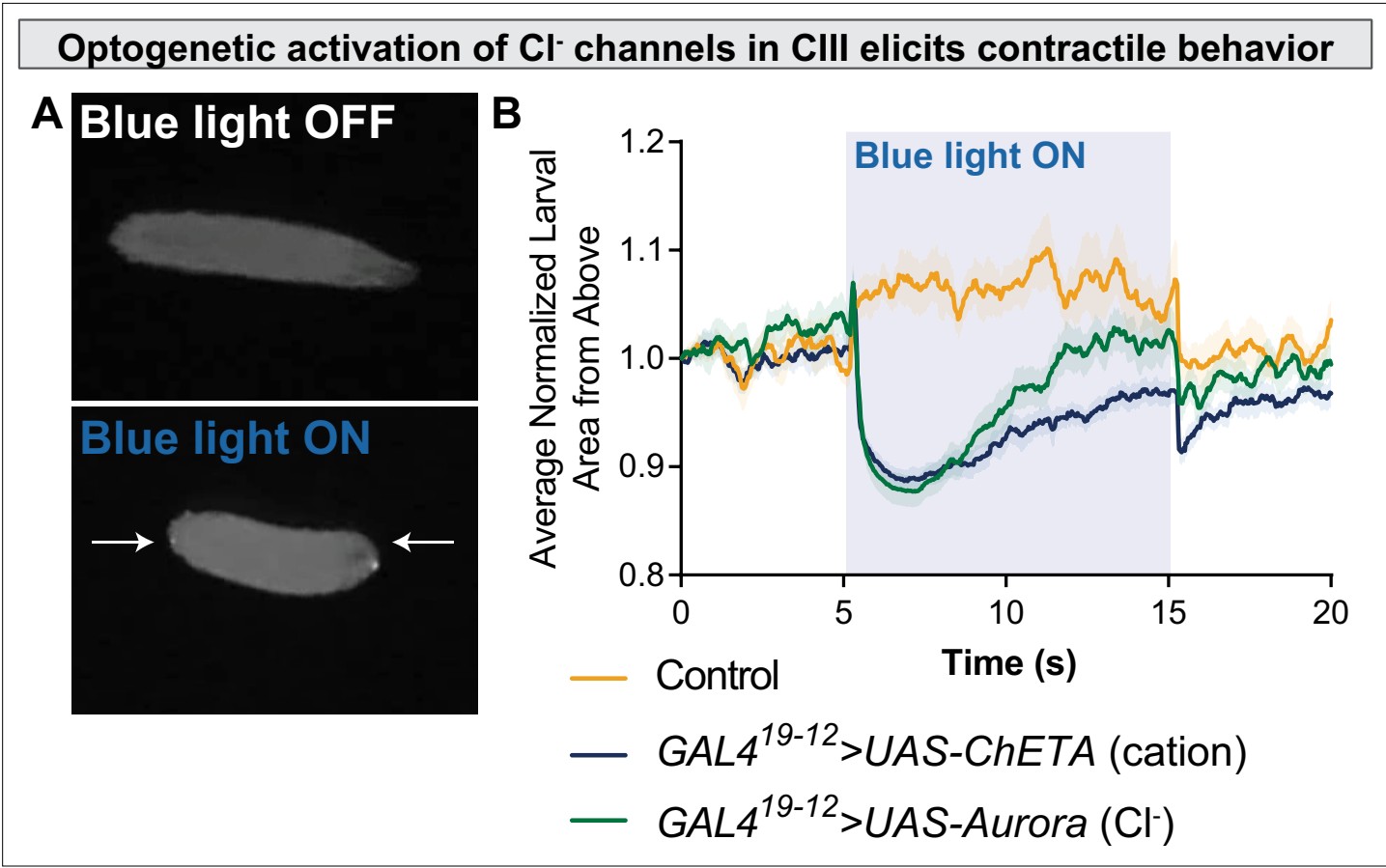

**Figure 5.** Optogenetic activation of chloride channels in CIII neurons elicits contractile behavior. (**A**) Optogenetic activation of class III (CIII) >Aurora activates CIII neurons, resulting in contraction (CT) behavior. (**B**) CT behavior represented as larval area from above. Blue light activation of the Ca$^{2+}$ channel ChETA and the Cl$^-$ channel Aurora causes a rapid reduction in normalized larval area from above, an indication of CT behavior.

The online version of this article includes the following video for figure 5:

**Figure 5—video 1.** Behavior elicited by blue light in a control larva.
https://elifesciences.org/articles/76863/figures#fig5video1

**Figure 5—video 2.** Behavior elicited by blue light in a larva expressing ChETA in class III neurons.
https://elifesciences.org/articles/76863/figures#fig5video2

**Figure 5—video 3.** Behavior elicited by blue light in a larvae expression Aurora in class III neurons.
https://elifesciences.org/articles/76863/figures#fig5video3

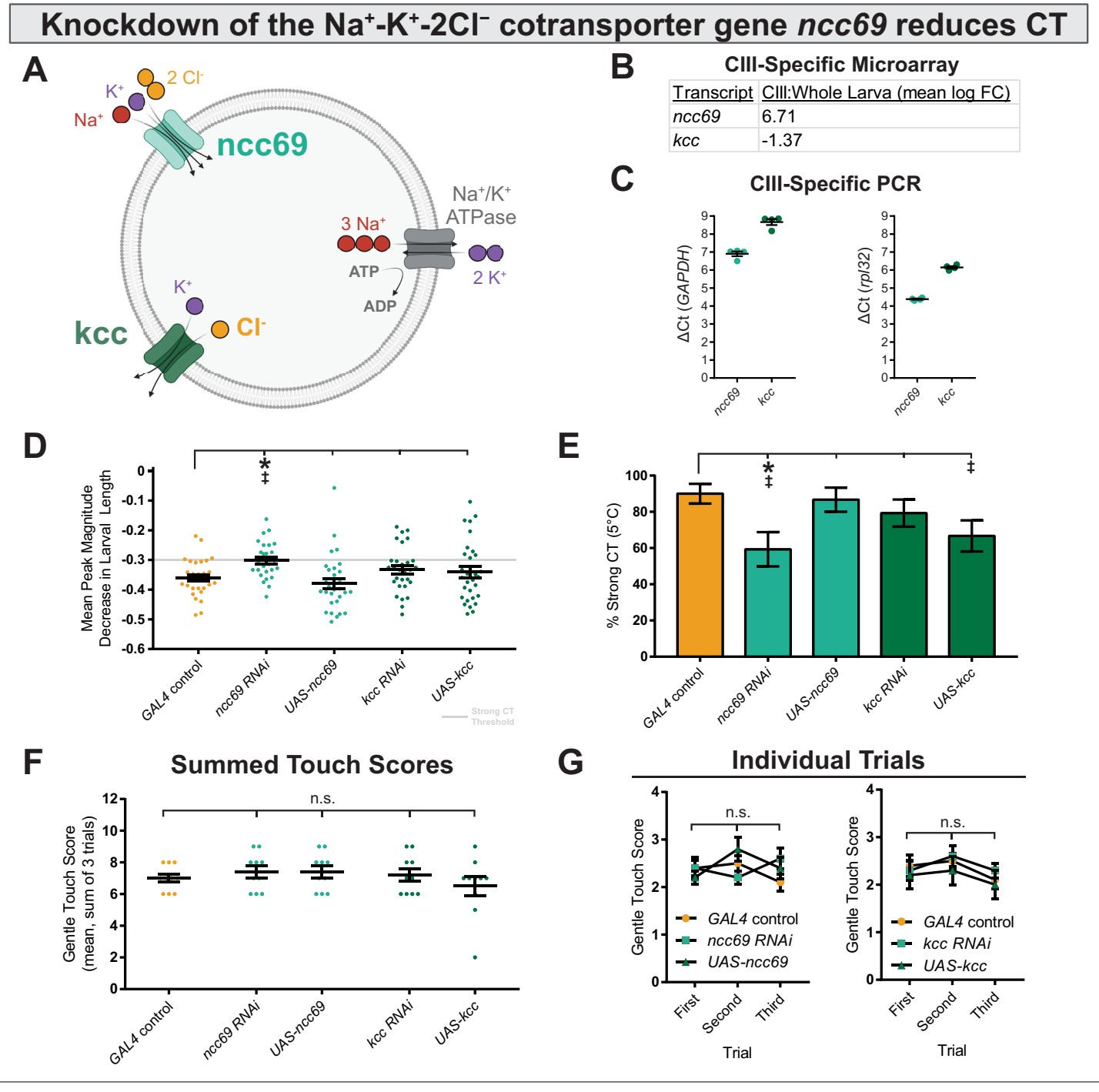

**Figure 6.** Genetically perturbing Cl⁻ homeostasis affects cold sensitivity, but not touch sensitivity. (**A**) Schematic of neural Cl⁻ homeostasis as modulated by the secondary active cotransporters ncc69 and kcc. (**B**) Class III (CIII) expression of *ncc69* and *kcc* from cell-type specific microarray (GSE69353), expressed as mean log fold-change difference between isolated CIII and whole-larval samples. Positive values indicate enrichment/upregulation and negative values downregulation. (**C**) qRT-PCR validating CIII expression of *ncc69* and *kcc* (n=4, each condition). (**D**) Mean peak magnitude in larval contraction. GAL4 control (n=30); *ncc69 RNAi* (n=29; p=0.025; BF₁₀=46.38); *ncc69 OE* (n=30; p=0.77; BF₁₀=0.38); *kcc RNAi* (n=29; p=0.51; BF₁₀=0.68); *kcc OE* (n=30; p=0.76; BF₁₀=0.37). (**E**) % of animals which strongly contract (CT) in response to noxious cold (≥30% reduction in body length). CIII-specific knockdown (*GAL4^19-12*) of *ncc69* results in a reduced percent of larvae which strongly CT in response to noxious cold. GAL4 control (n=30); *ncc69 RNAi* (n=29; p=0.038; BF₁₀=7.96); *ncc69 OE* (n=30; p=1; BF₁₀=0.58); *kcc RNAi* (n=29; p=0.91; BF₁₀=0.99); *kcc OE* (n=30; p=0.13; BF₁₀=3.29). (**F**) Average summed touch scores. (**G**) Average touch scores across trials.

The online version of this article includes the following figure supplement(s) for figure 6:

**Figure supplement 1.** Cold nociception defects of *ncc69* and *kcc* RNAi in CIII cold nociceptive neurons.

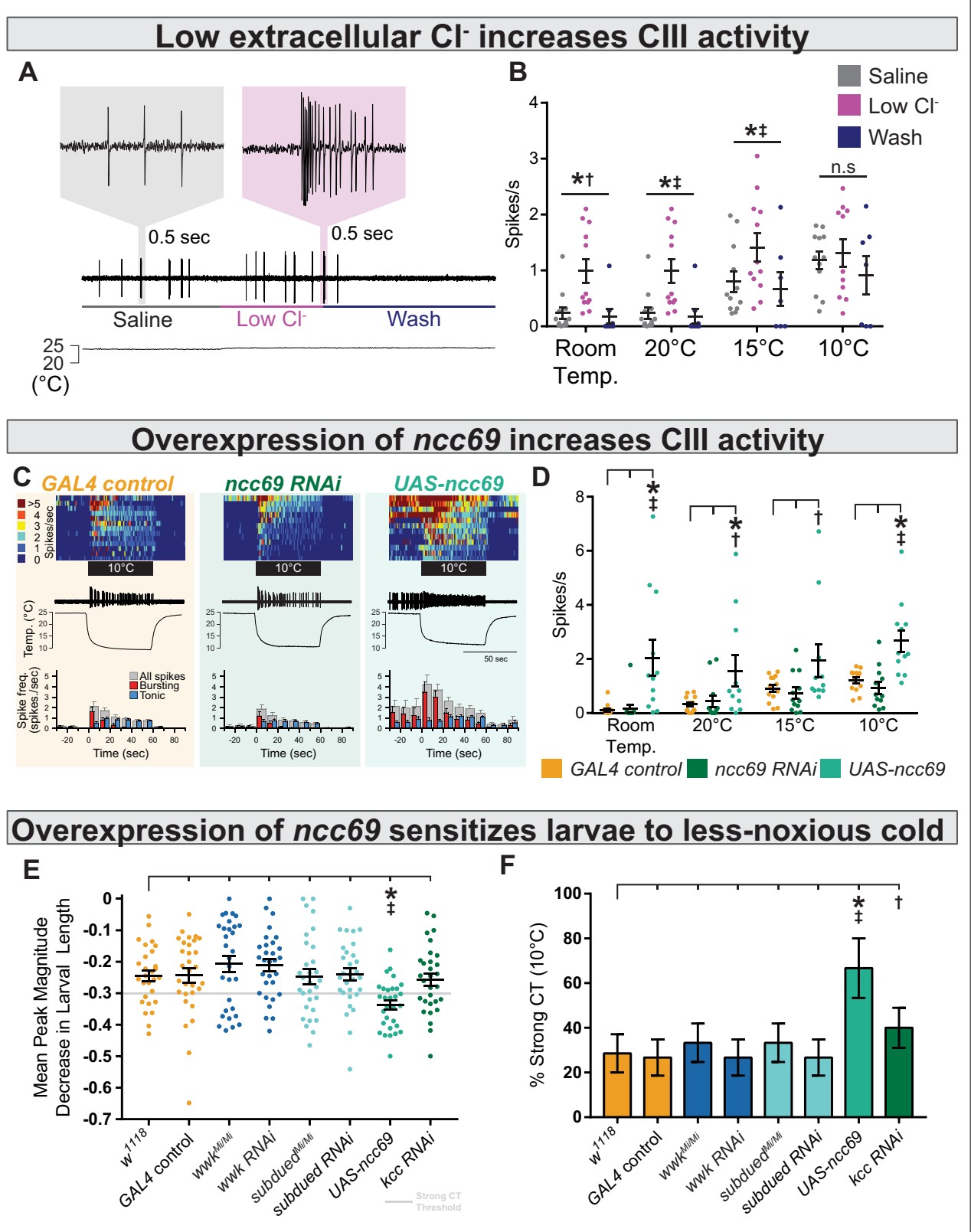

**Figure 7.** Decreasing extracellular Cl⁻ and overexpression of *ncc69* causes nociceptor and behavioral sensitization. (**A**) Extracellular application of low Cl⁻ saline to fileted electrophysiology preps causes spontaneous bursting activity. At room temperature, neurons are largely silent but may show occasional, low-frequency spiking (likely associated with mechanosensation from saline flow). (**B**) The presence of low Cl⁻ saline causes spontaneous firing and sensitizes class III (CIII) neurons to cooling. Room temp: saline (n=12); low Cl⁻ (n=12); wash (n=8); p=0.003, BF₁₀=1.89. 20°C: saline (n=12);

*Figure 7 continued on next page*

*Figure 7 continued*

low Cl$^-$ (n=12); wash (n=8); p=0.003, BF$_{10}$=66.38. 15°C: saline (n=12); low Cl$^-$ (n=12); wash (n=7); p=0.011, BF$_{10}$=14.51. 10°C: saline (n=12); low Cl$^-$ (n=11); wash (n=7); p=0.25, BF$_{10}$=0.75. (C) Top: Heatmap representation of cold-evoked CIII activity, with each line representing an individual sample prep. Middle: Representative traces of cold-evoked neural activity over graph of temperature ramp. Overexpression of ncc69 results in spontaneous nociceptor activity and increased cold sensitivity. Bottom: Representation of average spike frequency from population binned by 10 s. Red and blue bars show the proportion of bursting vs tonic spiking activity. (D) Overexpression of *ncc69* causes spontaneous neural activity and sensitizes neurons to cooling. Room temp: *GAL4* control (n=13); *ncc69* RNAi (n=12; p=0.99; BF$_{10}$=0.22); *UAS-ncc69* (n=12; p<0.001; BF$_{10}$=7.86). 20°C: *GAL4* control (n=13); *ncc69* RNAi (n=12; p=0.96; BF$_{10}$=0.40); *UAS-ncc69* (n=11; p=0.020; BF$_{10}$=2.23). 15°C: *GAL4* control (n=13); *ncc69* RNAi (n=12; p=0.90; BF$_{10}$=0.26); *UAS-ncc69* (n=11; p=0.062; BF$_{10}$=1.17). 10°C: *GAL4* control (n=13); *ncc69* RNAi (n=12; p=0.77; BF$_{10}$=0.60); *UAS-ncc69* (n=11; p=0.0042; BF$_{10}$=25.18). (E) Mean peak magnitude in larval contraction in response to noxious cold (10°C). *w$^{1118}$* (n=28); *GAL4 control* (n=30; p=1; BF$_{10}$=0.27); *wwk$^{Mi/Mi}$* (n=30; p=1; BF$_{10}$=0.48); *wwk RNAi* (n=30; p=1; BF$_{10}$=0.52); *subdued$^{Mi/Mi}$* (n=30; p=1; BF$_{10}$=0.27); *subdued RNAi* (n=30; p=1; BF$_{10}$=0.27); *UAS-ncc69* (n=30; p=0.037; BF$_{10}$=199.79); *kcc RNAi* (n=30; p=1; BF$_{10}$=0.29). (F) % of animals which strongly contract (CT) in response to noxious cold (10°C). CIII-specific (*GAL4$^{19-12}$*) overexpression of *ncc69* results in an increased percentage of strong CT in response to less-noxious cold. *w$^{1118}$* (n=28); *GAL4 control* (n=30; p=1; BF$_{10}$=0.46); *wwk$^{Mi/Mi}$* (n=30; p=1; BF$_{10}$=0.69); *wwk RNAi* (n=30; p=1; BF$_{10}$=0.46); *subdued$^{Mi/Mi}$* (n=30; p=1; BF$_{10}$=0.69); *subdued RNAi* (n=30; p=1; BF$_{10}$=0.46); *UAS-ncc69* (n=30; p=0.026; BF$_{10}$=28.79); *kcc RNAi* (n=30; p=1; BF$_{10}$=1.07).

The online version of this article includes the following figure supplement(s) for figure 7:

**Figure supplement 1.** Cold nociception defects of *subdued*, *white walker*, *ncc69*, and *kcc* at less-noxious cold temperature.

We next assessed cold nociception in response to less-noxious cold (10°C), which causes strong CT in only ~30% of wild-type animals and is not affected by *subdued* or *white walker* loss of function (*Figure 7E–F*, *Figure 7—figure supplement 1*). Based on our electrophysiological observations, we predicted that overexpression of *ncc69* and knockdown of *kcc* would result in cold hypersensitivity to less-noxious temperature. Consistent with these electrophysiological observations, *GAL4-UAS*-mediated overexpression of *ncc69* resulted in increased cold sensitivity at less-noxious temperatures; the magnitude of CT was substantially stronger (*Figure 7E*), and thus more subjects strongly CT in response to less-noxious cold (*Figure 7F*). However, unexpectedly, the cold-sensitivity phenotype was not clearly mirrored by *kcc* knockdown (a sensitization phenotype only barely more likely than no phenotype, as inferred by Bayesian analyses).

## Discussion

Here, we have shown that CIII cold nociceptors make use of excitatory Cl$^-$ currents to selectively encode cold. Our current working hypothesis in light of these findings is that cold-evoked, TRP-channel mediated Ca$^{2+}$ currents activate CaCCs, which due to differential expression of *ncc69* and *kcc*, result in depolarizing Cl$^-$ currents, enhancing neural activation in response to cold (*Figure 8*). These results support a role for *subdued*, *white walker*, and *ncc69* in selectively facilitating CIII-dependent cold nociception and not mechanosensation, thereby participating in mechanisms that allow CIII neurons to differentiate between sensory modalities. While our results provide strong evidence for Ca$^{2+}$-dependent mechanisms in the rapid response of CIII neurons to cooling, our studies also suggest that additional Ca$^{2+}$-independent mechanisms may also contribute to complex processes in these neurons that function in driving spiking activity, sensitization, and/or cold acclimation at colder temperatures.

As *subdued* has been previously characterized as a CaCC, its role is consistent with the hypothesis outlined above. However, the evolution of *subdued* has been implicitly debated in the literature (*Le et al., 2019*; *Wong et al., 2013*), with suggestion that it may be more closely related to ANO6 (*Le et al., 2019*). Our phylogenetic analysis strongly evidences that *subdued* is part of the bilaterian ANO1/ANO2 subfamily of CaCCs. Moreover, our phylogeny suggests that insects have no direct ANO6 homologue, as the diversification of ANO3, ANO4, ANO5, ANO6, and ANO9 occurred after the protostome-deuterostome split. The role of *subdued* in cold nociception therefore may constitute functional homology in the bilaterian ANO1/ANO2 subfamily, as mammalian ANO1 has been shown to participate in nociception alongside mammalian TRP channels (*Takayama et al., 2019*). However, the possibility of convergent evolution cannot yet be ruled due to the absence of evidence of function in other taxa.

In contrast, *white walker* has not been demonstrated to function as, or be closely related to, CaCCs. Our phylogeny evidences that *white walker* is part of the metazoan ANO8 subfamily; one important function of mammalian ANO8 is to tether the endoplasmic reticulum (ER) and plasma membrane

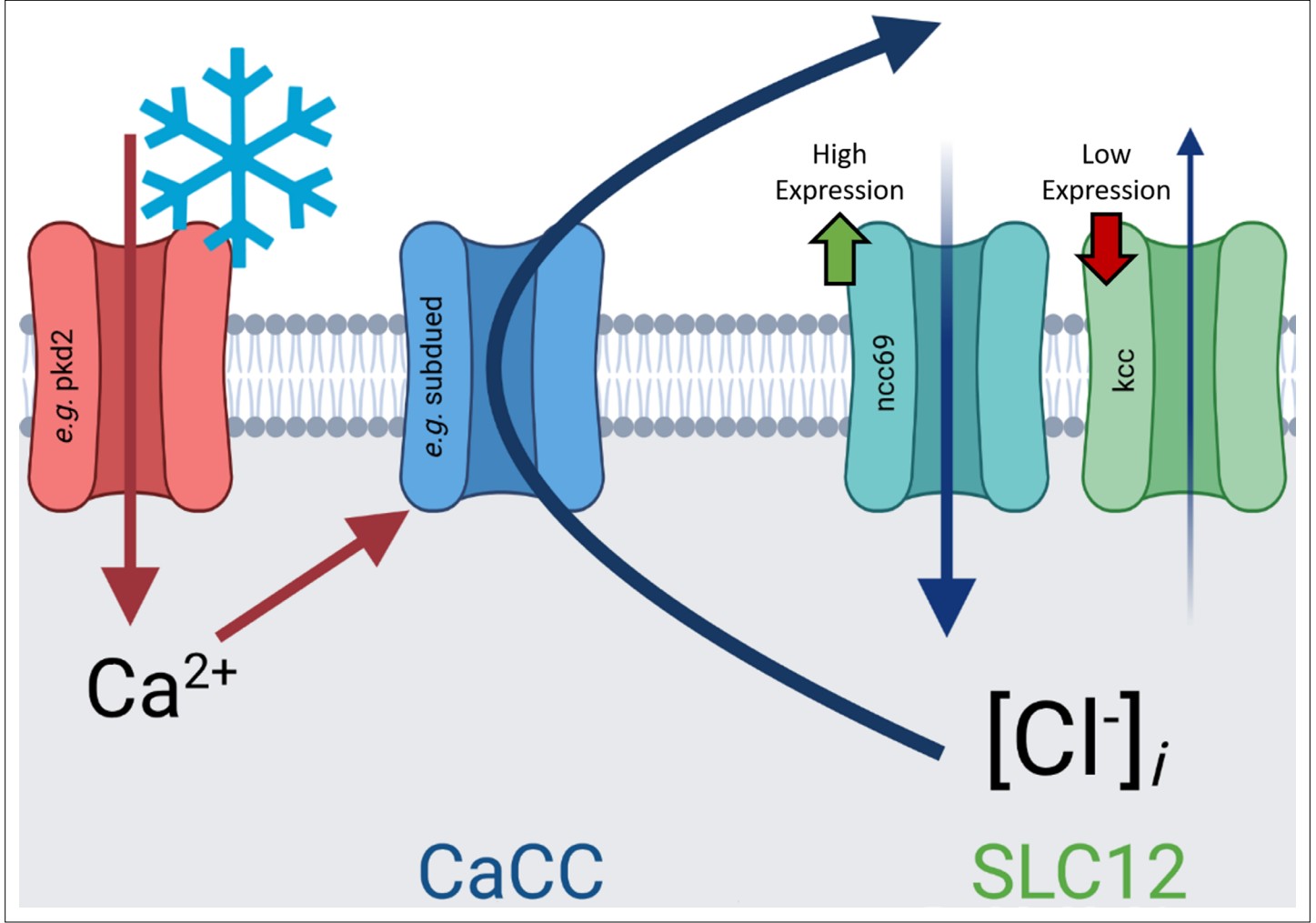

**Figure 8.** Graphical summary of hypothesis outlined in discussion: cold-evoked, transient receptor potential-channel mediated $Ca^{2+}$ currents activate $Ca^{2+}$-activated $Cl^-$ channels (CaCCs), which due to differential expression of *ncc69* and *kcc*, result in depolarizing $Cl^-$ currents, enhancing neural activation in response to cold.

(PM), thereby facilitating inter-membrane $Ca^{2+}$ signaling (*Jha et al., 2019*). Therefore, a speculative hypothesis is that *white walker* likewise serves to couple the ER and PM, and that subsequently, ER-dependent $Ca^{2+}$ signaling might promote the CIII cold response. In fact, a recent study (published contemporaneously with this study) has shown that ER-related $Ca^{2+}$-induced $Ca^{2+}$ release mechanisms are required for cold nociception (*Patel et al., 2022*). However, ANO8 has been shown to conduct $Cl^-$ heterologously (*Tian et al., 2012*), so $Cl^-$ channel function in *Drosophila* cannot be ruled out a priori. As *white walker* appears to be broadly expressed in neural tissues, *white walker* may function as a fundamental component of insect neural machinery and is therefore likely to be a gene of interest in future studies.

In addition to the functions outlined above, anoctamins—including subdued (*Le et al., 2019*)—are known to function as lipid scramblases (*Pedemonte and Galietta, 2014*). A plausible alternative hypothesis is therefore that *subdued* and/or *white walker* function as lipid scramblases as part of unidentified signaling cascades critical to noxious cold transduction.

The results of our *ncc69* knockdown behavior, $Cl^-$-channel optogenetics, and $Cl^-$ electrophysiology experiments are consistent with the hypothesis that CIII neurons make use of atypical excitatory $Cl^-$ currents. However, we did not observe an effect on cold-evoked CIII activity in response to *ncc69* knockdown (*Figure 7C–D*). It may be the case that this knockdown only affects electrical activity at very noxious temperatures; our inability to detect deficiencies in cold-evoked neural activity may therefore be due to limitations in our electrophysiology prep, which limit our ability to cool below 10°C. These

results are still curious; however, as *subdued* and *white walker* knockdowns result in electrophysiological defects at less-noxious (10°C) and innocuous (15°C) temperature drops (*Figure 3B–D*); despite this discrepancy, these electrophysiological differences do not correlate with behavioral differences at 10°C (*Figure 7E–F*) and decreased activity thus may not be behaviorally relevant at this temperature. Moreover, although we have substantial Bayesian evidence of an effect on % strong CT under *kcc* overexpression (*Figure 6E*), this difference was not evidenced by traditional frequentist statistics, the phenotype did not clearly mimic *ncc69* knockdown (*Figure 6E*), nor did we see a difference in the mean peak magnitude of CT response (*Figure 6D*). In totality, these results may suggest that CIII Cl⁻ homeostasis involves other cotransporters which can adapt in either function or expression in response to loss or gain of function. Given the importance of this system to behavior selection, CIII Cl⁻ homeostasis will make an interesting target for future experimentation.

Importantly, we have shown that overexpression of *ncc69*—a fly orthologue of *NKCC1*—is sufficient for driving a sensitization of cold nociception in larvae. As dysregulation of *NKCC1* and *kcc* is associated with neuropathic pain in mammals (*Mòdol et al., 2014*), we posit that altered larval cold nociception constitutes a new system in which to study neuropathic pain. Importantly, this system is wholly genetic and does not require injury or other methods of invoking nociceptive sensitization, making it a high throughput and easily accessible tool. Interestingly, RNAi knockdown of *kcc* did not mirror the *ncc69* overexpression phenotype (*Figure 7E–F*). We speculate that this is because native *kcc* expression levels are low enough that knockdown does not sufficiently disrupt Cl⁻ homeostasis. This might also be because of hypothetical unknown mechanisms of compensation, as discussed above.

While it has been often stated that neuropathic pain is maladaptive, there is growing support for the hypothesis that neuropathic pain has its mechanistic bases in adaptive nociceptive sensitization—a mechanism by which organisms are more readily able to respond to danger following insult (*Crook et al., 2014*; *Descalzi et al., 2017*; *Gasull et al., 2005*; *Géranton, 2019*; *Howard et al., 2019*; *Khuong et al., 2019*; *Nesse and Schulkin, 2019*; *Perrot-Minnot et al., 2017*; *Price and Dussor, 2014*; *Takayama et al., 2019*; *Walters, 2019*; *Walters and Williams, 2019*; *Williams, 2019*). Nerve injury has been previously shown to cause nociceptive sensitization in adult *Drosophila* and has been hypothesized to be protective (*Khuong et al., 2019*). Moreover, it has been recently shown that hyperexcitability of CIII neurons is coincident with cold acclimation, the mechanism by which insects adapt to dips in temperature (*Himmel et al., 2021*). One speculative hypothesis is that changes in expression levels of SLC12 transporters underlie these shifts in cold acclimation-induced cold sensitivity. This would be consistent with a study demonstrating that a number of genes in *Drosophila* involved in ion homeostasis are differentially regulated following cold acclimation (*MacMillan et al., 2016*). If this speculation is veridical, insect thermal acclimation may serve as an example of how 'maladaptive' injury and neuropathic sensitization can confer an adaptive advantage. It is therefore possible that these findings, and continued study, will lead to not only advances relevant to human health but also better our understanding of nervous system evolution and the evolution of mechanisms underlying neuropathic sensitization and pain.

## Materials and methods

### Fly strains

All *D. melanogaster* stocks were maintained at 24°C under a 12:12 light cycle. Genetic crosses were raised at 29°C under a 12:12 light cycle in order to accelerate development. Third instar larvae were used for all experiments. Publicly available transgenic strains were originally sourced from the Bloomington *Drosophila* Stock Center (B) and the Vienna *Drosophila* Resource Center (v) and included: *GAL4^nompC* (B36361); *T2A-GAL4^wwk/CG15270* (B76649); *UAS-Aurora* (B76327); *UAS-mCD8::GFP* (B5130); *wwk^MI03516* (B36976); *Df(2 L)b87e25* (*wwk df*, B3138); *subdued^MI15535* (B61082); *Df(3 R)Exel6184* (*subdued df*, B7663); *UAS-subdued RNAi* (#1, v37472; #2, v108953); *UAS-wwk RNAi-1* (#1, B28650; #2, B62282); *UAS-ncc69 RNAi* (B28682); and *UAS-kcc RNAi* (B34584). *UAS-kcc* was provided by Dr. Mark Tanouye; *UAS-ncc69* was provided by Dr. Don van Meyel; *UAS-subdued* and *GAL4^c240* were provided by Dr. Changsoo Kim. *GAL4^19-12* is available upon request.

*CIII::tdTomato* and *UAS-wwk* were developed for this study and are available upon request. For *CIII::tdTomato,* we utilized Gateway cloning to combine the *R83B04* (CIII) enhancer with *CD4-tdTomato*

by LR reaction (Invitrogen ref: 12538–120). Transgenic flies carrying a second chromosomal insertion of *CIII:tdTomato* (docking site: VK37) were generated by Genetivision. The *R83B04* enhancer-containing entry vector was provided by the FlyLight Project. *pDEST-HemmarR2*—the vector for enhancer driven *CD4-tdTomato* expression—was provided by Dr. Chun Han. R83B04 was identified in a screen seeking to identify new CIII neuron drivers (data not shown; also see *Patel et al., 2022* for other uses). We first visually screened publicly available larval CNS expression patterns of GAL4s from the FlyLight Project. Next, we performed an optogenetic/behavioral screen on select candidate GAL4s and identified GMR83B04 as a CIII driver (here, *GAL4^CIII*). CIII tagging in the R83B04-driven *CIII::tdTomato* construct was verified by visualizing overlapping expression patterns with the previously identified *GAL4^nompC* (*Figure 1—figure supplement 1*).

For *UAS-wwk*, full-length cDNA was synthesized (GenScript) and subcloned into *pUAST-attB*. Transgenic *UAS-wwk* flies carrying a third chromosomal insertion of *UAS-wwk* (docking site: VK20) were generated by Genetivision.

All biological materials/strains are available upon request to the corresponding author.

### Polymerase chain reaction

For PCR on isolated CIII samples, isolation of CIII neurons followed a previously described protocol (*Iyer et al., 2009*). In brief, 40–50 third instar larvae with mCD8::GFP-tagged CIII neurons were collected and washed in ddH$_2$0, dissected using microdissection scissors, and then dissociated in PBS using a glass dounce. CIII neurons were then isolated using superparamagnetic beads (Dynabeads MyOne Steptavidin T1, Invitrogen) that were conjugated to biotinylated anti-mCD8a antibody (eBioscience). RNA was isolated from these neurons using the RNeasy Mini Kit (Qiagen), and qRT-PCR analysis was performed using pre-validated Qiagen QuantiTect Primer Assays for *CG15270* (QT00936754), *subdued* (QT00978131), *ncc69* (QT00963263), *kcc* (QT00953862), and normalized against *GAPDH* (QT00922957), and *rpl32* (QT00985677).

For whole larval samples, age-matched *Drosophila* homozygous larvae were collected from *w^1118*, *subdued^MI15535*, and *wwk^MI03516*, and total RNA was extracted from whole larvae using RNeasy Mini Kit (Qiagen). For RT-PCR, input RNA was identical across genotypes, and the following primer sequences were used: *subdued*-forward: 5′-CCA CAA TCC CGC CAT ATC A-3′; *subdued*-reverse: 5′-GTA GGC CAG GAC AAA GTC AA-3′; *wwk*-forward: 5′-ACA ACG GCG ACT TCA ACA-3′; *wwk*-reverse: 5′-CTG GTG CAT CCT CAG GAA AT-3′; RT-PCR was performed using the one-step RT-PCR kit (Qiagen). The *subdued* primers are located in exons 1 and 2 and are predicted to produce an amplicon of 499 base pairs, while the *wwk* primers are located in exons 1 and 2 and are predicted to produce an amplicon of 271 base pairs. All primers were obtained from IDTDNA.

### Cold nociception assay

Third instar larvae were raised in and collected from standard fly food vials, gently washed in water, then acclimated to a room temperature, moistened, black-painted aluminum arena. Once locomotion resumed, the arena was transferred to a prechilled Peltier device (TE Technologies, CP0031) under the control of a thermoelectric temperature controller (TE Technologies, TC-48–20). Behavior was recorded from above by a Nikon D5200 DSLR camera, and the first 5 s of footage following contact between the arena and Peltier plate were used for behavioral analysis. Videos were converted to the avi format by Video to Video software (https://www.videotovideo.org/) and then further processed in the FIJI distribution of ImageJ. Virtual stack images of larvae were converted to grayscale, thresholded, and subsequently skeletonized into single pixel-wide lines, representing larval length from tip to tip. Length measurements were normalized to the length at time 0, and CT was counted if the larvae passed the CT threshold at any point during the analysis. The CT threshold was determined as in previous studies, as the average peak *w^1118* decrease in length + 1.5 × the standard deviation; this resulted in a CT threshold (here called strong CT) of approximately 0.7 (also presented as –0.3 change in normalized body length or 30% reduction in body length). Data are reported as % CT and the peak magnitude of decrease in larval length. Some larval behavior recordings were discarded because of visual noise generated during the thresholding process, making accurate measures of larval length impossible.

### Mechanosensation assay

Third instar larvae were collected, gently washed, then acclimated to the same arena used in cold plate assays. Mechanosensitivity was scored similarly to previous reports (*Kernan et al., 1994*; *Turner*

*et al., 2016*): in brief, animals were brushed on an anterior segment by a single paintbrush bristle, and behavior was observed through a Zeiss Stemi 305 microscope. Innocuous touch behaviors include pausing/hesitating, head/anterior withdrawal (AKA hunching), head casting/turning, and reverse locomotion. Each subject was given one point if it performed one of these behaviors (for a maximum of four points per trial). Each animal was subject to three touch trials with 30 s intervals between each trial. The scores from the three trials were summed (for a maximum of 12 points per subject) and averaged across each genotype.

## Locomotion assay

Locomotion of third instar larvae was assessed on agarose gel and quantitively analyzed using FimTrack (*Risse et al., 2017*). The assay was performed as described in *Reddish et al., 2021*. Larvae were rinsed and placed on 5% agarose gel and allowed to acclimate for 5 min. The agarose gel was illuminated from below, and larval locomotion was recorded from above using a Nikon D5300 DSLR camera at 30 frames per second. Larval locomotion was recorded for 5 min. Raw locomotion videos were processed using Video to Video software (https://www.videotovideo.org/) and the Fiji distribution of ImageJ. Quantitative analysis of larval locomotion was performed using FimTrack (*Risse et al., 2017*). Locomotion tracking can be impaired if larvae locomote out of field of view or collide with other larvae; therefore, we analyzed only continuous 1-min-long locomotion tracks, where individual larvae did not collide with one another.

## Optogenetics

Third instar larvae were collected, gently washed, then acclimated to a glass plate. The plate was then transferred to a custom-built optogenetics-behavior rig fitted with bottom-mounted blue LED illumination and a top-mounted Canon Rebel T3i DSLR camera. Blue light, video recording, and larval tracking were handled automatically by Noldus EthoVision software; behavior was recorded for 5 s in the absence of blue light, for 10 s with blue light illumination, and for 5 additional seconds in the absence of blue light. Larval behavior was quantified by automated tracking of larval area from above.

## Microscopy

For morphometric analyses, GFP tagged CIII neurons in third instar larvae were imaged on a Zeiss LSM 780 confocal microscope at 200✕ magnification. Images were collected as z-stacks with a step size of 2.0 µm and at 1024 × 1024 resolution. Maximum intensity projections of z-stacks were exported using Zen (blue edition) software and analyzed using the Analyze Skeleton ImageJ plugin. Metrics were compiled using custom Python algorithms freely available upon request.

To image *GAL4* expression patterns, larvae were likewise imaged under Zeiss LSM 780 confocal microscope at various magnifications (scale bars present on relevant images), at various locations (indicated in relevant figures/legends). Images were viewed and exported from Zen (black edition) software and labeled in Adobe Illustrator.

## Electrophysiology

Demuscled fillet preparations made from third instar larvae were placed on the bottom of an experimental chamber (200 µL) filled with HL-3 saline, which consisted of (in mM) 70 NaCl, 5 KCl, 1.5 CaCl$_2$, 20 MgCl$_2$, 10 NaHCO$_3$, 5 trehalose, 115 sucrose, and 5 HEPES (pH 7.2). For BAPTA-AM, the compound was perfused at a concentration of 100 µM. For the low-chloride saline, sodium chloride and magnesium chloride were replaced with sodium sulfate (Na$_2$SO$_4$, 35 mM) and magnesium sulfate (MgSO$_4$, 20 mM). The osmolarity of the low-chloride saline was adjusted to become equal to HL-3 saline (355–360 mOsm) by adding sucrose (up to 170 mM in total). The specimen was constantly perfused with HL-3 saline (30–40 µL/s). Switching between HL-3 and low-chlorine saline was done by a three-way stopcock attached to the inlet tube.

Cold temperature stimulation was administered by passing the saline through an in-line solution cooler (SC-20, Warner Instruments, Hamden, CT, USA) connected to the controller device (CL-100, Warner Instruments, Hamden, CT, USA). The saline temperature was constantly monitored by a thermometer probe (BAT-12, Physitemp, Clifton, NJ, USA) placed adjacent to the preparation.

The spiking activity of a GFP-tagged CIII neurons was recorded extracellularly by using a borosilicate glass micropipette (tip diameter, 5–10 µm) by applying gentle suction. The electrode was connected

to the headstage of a patch-clamp amplifier (MultiClamp 700 A, Molecular Devices, San Jose, CA, USA). All recordings were made from ddaA in the dorsal cluster of the peripheral sensory neurons. The amplifier's output was digitized at 10-kHz sampling rate using a Micro1401 A/D converter (Cambridge Electronic Design, Cambridge, UK) and acquired into a laptop computer running Windows 10 (Microsoft, Redmont, WA, USA) with Spike2 software v. 8 (Cambridge Electronic Design, Cambridge, UK). Bursting spikes were identified as those with inter-spike intervals of less than 0.2 s.

## Phylogenetics

Starting with previously characterized ANO sequences from human (NCBI CCDS), mouse (NCBI CCDS), and *D. melanogaster* (FlyBase), an ANO sequence database was assembled by Blastp against protein sequence/model databases at Ensembl and the Okinawa Institute of Science and Technology Marine Genomes Unit. In order to maximize useful phylogenetic information, only Blast hits >300 amino acids in length with an E-value less than 1E−20 were retained.

Next, CD-HIT was used to identify and remove duplicate sequences and predicted isoforms (threshold 90% similarity), retaining the longest isoform to maximize phylogenetic information. Phobius was then used to predict TM topology to further maximize phylogenetic information, sequences with fewer than five predicted TM segments were removed.

Sequences were thereafter aligned with a transmembrane channel-like outgroup via MAFFT using default settings. The final phylogenetic tree was generated via IQ-Tree by the maximum likelihood approach, using an LG +R7 substitution model (as determined by ModelFinder). Branch support was calculated by ultrafast bootstrapping (UFboot, 2000 replicates). To identify duplication events, the maximum likelihood phylogeny was reconciled using NOTUNG 2.9.1. To formulate the most parsimonious interpretation of the resulting trees, weakly supported branches were rearranged (UFboot 95 cutoff) in a species-aware fashion against the species cladogram in *Figure 4A*; edge weight threshold was set to 1.0, and the costs of duplications and losses were set to 1.5 and 1.0, respectively. Trees were visualized using iTOL and Adobe Illustrator.

## Statistical analyses and data availability

Due to a growing call for statistical analyses that do not rely on p-values (*Ioannidis, 2019*; *Kelter, 2020*; *Keysers et al., 2020*; *Matthews et al., 2017*; *Nuijten et al., 2016*; *van Doorn et al., 2021*; *Wasserstein and Lazar, 2016*; *Wasserstein et al., 2019*; *Wetzels et al., 2011*), differences in mean and population proportions were analyzed using both traditional frequentist statistics and Bayesian alternatives.

Population proportions are presented as % ± standard error of the proportion; differences in proportion were assessed by one-tailed (for 5°C experiments) or two-tailed (for 10°C experiments) z-test with a Bonferroni correction and the Bayesian A/B test. All other measures are presented as mean ± standard error of the mean unless otherwise noted; differences were assessed by frequentist one- or two-way ANOVA with Dunnett's post-hoc tests and Bayesian equivalents (*Westfall et al., 1997*), or frequentist and Bayesian t-test (in the case of comparisons between only two groups). Electrophysiology low Cl⁻ wash experiments were performed within-subjects. For some subjects, recordings were lost during the low Cl⁻ or washout phase; to account for these missing data, differences were assessed using a linear mixed model and Bayesian repeated measures ANOVA (both analyzing the effect of different washes on firing frequency). z-Tests were performed in Microsoft Excel, two-way ANOVA in GraphPad PRISM, and all other analyses in JASP (Bayesian analyses using default prior probability distributions or priors adjusted by the Westfall-Johnson-Utts method) (*Team, J, 2020*).

For frequentist analyses, statistical significance was assessed at $\alpha$=0.05. For Bayesian analyses, degree of support for rejecting the null hypothesis was inferred by computed Bayes factors (BF), via a modification on the method originally proposed by *Jeffreys, 1961*: 'null hypothesis supported' ($BF_{10}$ <1); 'weak' ($BF_{10}$: 1–3); 'substantial' ($BF_{10}$: 3–10); 'strong' ($BF_{10}$: 10–30); 'very strong' ($BF_{10}$: 30–100); 'decisive' ($BF_{10}$ >100).

The datasets supporting this study are reported in the manuscript, and raw experimental data and sequence data for phylogenetic analyses are available from Dryad. No statistical methods were used to predetermine sample size; sample sizes were determined based on prior experimentation and are consistent with other behavioral studies in *Drosophila*. The findings described here were tested via several complementary methods and genotypes, as biological replicates from different populations

of animals, over several days of experimentation, to ensure reproducibility. Subjects were chosen at random from populations housed in individual vials separated by genotype. Experimenters were not blinded to genotypes or any other conditions in these experiments.

All graphs were generated using GraphPad PRISM (GraphPad Software, La Jolla, California, USA) and Adobe Illustrator. Relevant n, adjusted p, and $BF_{10}$ values are listed in figure legends. In figures, asterisks (*) indicate statistical significance at $\alpha=0.05$; daggers (†) indicate weak evidence in favor of the alternative hypothesis; double daggers (‡) indicate at least substantial evidence in favor of the alternative hypothesis; and families of comparisons are grouped by overhead bars.

## Acknowledgements

This work is supported by NIH R01 NS115209-01 (to DNC and GSC). NJH was supported by NIH F31 NS117087-01, a GSU Brains & Behavior Fellowship, and a Kenneth W and Georganne F Honeycutt Fellowship. AAP was supported by a GSU Brains & Behavior Fellowship, a GSU 2 CI Neurogenomics Fellowship, and a Kenneth W and Georganne F Honeycutt Fellowship. JML was supported by a GSU Brains & Behavior Fellowship. We thank Dr Don van Meyel (McGill University), Dr Mark Tanoye (UC Berkeley), Dr Changsoo Kim (Chonnam National University), Dr Chun Han (Cornell University), and the FlyLight Project (Janelia Research Campus) for providing *Drosophila* stocks.

## Additional information

### Funding

| Funder | Grant reference number | Author |
|---|---|---|
| National Institutes of Health | NS115209 | Gennady S Cymbalyuk Daniel N Cox |
| National Institutes of Health | NS117087 | Nathaniel J Himmel |
| Georgia State University | Honeycutt Fellowship | Nathaniel J Himmel Atit A Patel |
| Georgia State University | Brains & Behavior Fellowship | Nathaniel J Himmel Atit A Patel Jamin M Letcher |

The funders had no role in study design, data collection and interpretation, or the decision to submit the work for publication.

### Author contributions

Nathaniel J Himmel, Conceptualization, Formal analysis, Funding acquisition, Investigation, Visualization, Methodology, Writing - original draft, Writing - review and editing; Akira Sakurai, Atit A Patel, Shatabdi Bhattacharjee, Formal analysis, Investigation, Visualization, Methodology, Writing - review and editing; Jamin M Letcher, Investigation, Visualization, Writing - review and editing; Maggie N Benson, Thomas R Gray, Investigation, Writing - review and editing; Gennady S Cymbalyuk, Funding acquisition, Visualization, Writing - review and editing; Daniel N Cox, Conceptualization, Supervision, Funding acquisition, Methodology, Writing - review and editing

### Author ORCIDs

Nathaniel J Himmel ⓘ http://orcid.org/0000-0001-7876-6960
Akira Sakurai ⓘ http://orcid.org/0000-0003-2858-1620
Jamin M Letcher ⓘ http://orcid.org/0000-0003-3077-0615
Thomas R Gray ⓘ http://orcid.org/0000-0002-6176-005X
Gennady S Cymbalyuk ⓘ http://orcid.org/0000-0002-1889-5397
Daniel N Cox ⓘ http://orcid.org/0000-0001-9191-9212

### Decision letter and Author response

Decision letter https://doi.org/10.7554/eLife.76863.sa1
Author response https://doi.org/10.7554/eLife.76863.sa2

## Additional files

### Supplementary files
• Transparent reporting form

### Data availability
All data generated or analyzed during this study are included in the manuscript. We have plotted all data such that individual data points can be seen. Additionally, we have included heatmap representations of cold-evoked larval behavior for every single larva used in this study (see figure supplements). Class III neuron-specific transcriptomic data has been deposited in GEO under accession number GSE69353. Accession numbers for sequences used in phylogenetic analysis are presented in the relevant figure. Raw data and sequence files are available in Dryad.

The following dataset was generated:

| Author(s) | Year | Dataset title | Dataset URL | Database and Identifier |
|---|---|---|---|---|
| Himmel NJ, Sakurai A, Patel A, Bhattacharjee S, Letcher J, Benson M, Gray T, Cymbalyuk G, Cox DN | 2022 | Choride-dependent mechanisms of multimodal sensory discrimination and nociceptive sensitization in *Drosophila* | http://dx.doi.org/10.5061/dryad.dncjsxm3h | Dryad Digital Repository, 10.5061/dryad.dncjsxm3h |

The following previously published dataset was used:

| Author(s) | Year | Dataset title | Dataset URL | Database and Identifier |
|---|---|---|---|---|
| Iyer SC, Bhattacharya S, Cox DN | 2016 | Microarray gene expression profiling of isolated class III Drosophila dendritic arborization sensory neurons | https://www.ncbi.nlm.nih.gov/geo/query/acc.cgi?acc=GSE69353 | NCBI Gene Expression Omnibus, GSE69353 |

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
