## [Editor Report]

This is an important manuscript that clarifies mechanisms of multimodality in a class of insect somatosensory neurons and presents a model for how Cl^-^ currents underlie cold nociception. The authors support the claims in the paper through the use of gene knockdown, behavioral experiments, neuroanatomy, and optogenetic activation in the *Drosophila* fruit fly. The demonstration that the same class of somatosensory neurons can respond to innocuous versus noxious stimuli, depending upon which protein those neurons are activated, could shed light on disease states such as neuropathic pain when touch signals are confused as painful.

---

## [Decision Letter]

**Decision letter after peer review:**

Thank you for submitting your article "Chloride-dependent mechanisms of multimodal sensory discrimination and neuropathic sensitization in *Drosophila*" for consideration by *eLife*. Your article has been reviewed by 3 peer reviewers, one of whom is a member of our Board of Reviewing Editors, and the evaluation has been overseen K VijayRaghavan as the Senior Editor. The following individual involved in the review of your submission has agreed to reveal their identity: Edgar T Walters (Reviewer #3).

Essential revisions:

All three reviewers are convinced of the high interest this study will generate on the important topic of polymodal sensory discrimination. Nonetheless, the authors should focus their attention on these three topics to improve the manuscript:

1) The authors should use their physiological preparation to test ca^2+^ dependence of anoctamin activation by cold stimulation.

2) Reviewers had concerns about the mutants that were used (not characterized other than location of insertion) and expression analysis that they show, especially for subdued. Enhancer traps on their own are very weak evidence for expression patterns, often misleading.

3) The behavioral analyses could benefit from the preponderance of new automated behavioral tools to get at richer phenotypic profiling across their knockdown experiments.

*Reviewer #3 (Recommendations for the authors):*

A specific suggestion for testing the proposed role for ca^2+^ in the model in Figure 8 would be to put BAPTA-AM into the recording pipette and/or bath, which should reduce cold-induced activity if ca^2+^-activated Cl^-^ conductances are important for this response.

The mentions of mammalian neuropathic pain papers related to spinal cord injury (SCI) on p.18 and elsewhere are not germane because the cited studies examined spinal neurons rather than primary sensory (DRG) neurons, they failed to describe "spontaneous nociceptor activity," and they described results from peripheral neuropathic models rather than SCI models. More appropriate would be references to peripheral neuropathic pain models that examined DRG neurons (e.g., PMID 24813295 and PMC6990784).

p.3, line 57 – There appears to be a typo. What does "variably selection" mean?

p.4, line 78 – Again, there is little evidence for or against upregulation of NKCC1 in sensory neurons contributing to human neuropathic pain after SCI.

p.10, Figure 2 headings – The data presented show that anoctamins contribute to the effects shown, but not that they are "required" for either CT responses or electrical activation by cold.

p.12, Figure 3A - The y-axis label reads "trails" rather than "trials".

p.15, line 256 – For clarity the sentence should add "compared to the average expression levels for the whole body," or something equivalent.

p.15, line 274 – The first sentence should read "… selectively required for responses to cold stimulation." The next sentence would be clearer if it read "… alternative hypothesis that Cl^-^ currents in CIII neurons are inhibitory but have excitatory downstream effects; for example, …"

p.16, line 286 – The alternative hypothesis is not disproved, but the Aurora findings do provide strong evidence against that hypothesis.

p.18, lines 310+ … – See weakness #5 above.

p.20, Figure 7 – The phrase "causes CIII neuropathic sensitization" in the heading is incorrect because there is no reason to assume that overexpression of ncc69 causes neuropathy. What is shown is neuronal sensitization. A better title would be "Overexpression of ncc69 sensitizes and induces ongoing activity in CIII neurons."

p.24, line 419 – References should be cited for this claim.

---

## [Author Response]

Essential revisions:All three reviewers are convinced of the high interest this study will generate on the important topic of polymodal sensory discrimination. Nonetheless, the authors should focus their attention on these three topics to improve the manuscript:1) The authors should use their physiological preparation to test ca^2+^ dependence of anoctamin activation by cold stimulation.

As suggested by Reviewer #3, we have performed an electrophysiological experiment using BAPTA-AM to chelate calcium. As predicted by the reviewer, BAPTA-AM reduced cold-evoked activity in CIII neurons. Although, we caution that BAPTA-AM is likely having an effect on all calcium-dependent cellular mechanisms.

2) Reviewers had concerns about the mutants that were used (not characterized other than location of insertion) and expression analysis that they show, especially for subdued. Enhancer traps on their own are very weak evidence for expression patterns, often misleading.

Firstly, we have performed RT-PCR to validate the mutant lines. These data can be found in Figure 2—figure supplement 1. We hope the use of PCR, MiMIC, chromosomal deficiency, transheterozygotes, CIII-specific rescue, and CIII-specific RNAi knockdown (for both behavior and electrophysiology) now sufficiently evidence that these genes are required in CIII for cold nociception.

To address the reviewers’ concerns re: expression, we have repeated the Class III specific qPCR experiments against 2 housekeeping genes (GAPDH and rpl32) and moved the data from the text to a graph in Figure 1. We have also moved the microarray analysis to Figure 1 to make it clearer to a reader that this was the method first used to identify CIII expression of subdued and wwk, and to better present that the GAL4 expression patterns are only supporting evidence. With respect to SLC12 cotransporters, we have added corroborating qPCR evidence of their expression (Figure 6). We want to strongly emphasize that the comparison between the two genes via qPCR is only semi-quantitative; however, the DCt value for *ncc69* was lower than *kcc*. Assuming similar PCR efficiency this would indicate more *ncc69* transcripts were present in the isolated Class III samples.

3) The behavioral analyses could benefit from the preponderance of new automated behavioral tools to get at richer phenotypic profiling across their knockdown experiments.

The cold-evoked behaviors were recorded so that they could be analyzed via the described ImageJ/FIJI pipeline – we do not believe the videos can be readily adapted to other tools. While there are now a series of publications differently exploring the intricacies of cold-evoked behavior (Turner and Armengol *et al.* 2016 https://doi.org/10.1016/j.cub.2016.09.038; Turner and Patel *et al.* 2018 https://doi.org/10.1371/journal.pone.0209577; Himmel et al. 2021 https://doi.org/10.1016/j.isci.2021.102657; and most recently, Patel *et al.* 2022 https://doi.org/10.3389/fnmol.2022.942548), the predominant cold-evoked behavior, in *D. melanogaster* and across a broad range *Drosophila* species, is simply cessation of locomotion and decrease in larval length (Himmel *et al.* 2021 https://doi.org/10.1016/j.isci.2021.102657), which is captured by the present behavioral heatmaps, length quantification, and CT categorization. Our group is continuing to study the intricacies of cold-evoked behavior and making use of new tools in our continued studies, but we do not have the new tools to add to this manuscript.

We have, however, made several additions/changes to this study in the spirit of the reviewers’ suggestions:

We have used the present data to plot a “failure to maintain contraction” metric, wherein we use our measures of length to count the number of animals that cease any contractile behavior (*i.e.* return to/surpass their original length). Please see supplemental figures for each main piece of behavioral data.We have made two changes to better show that % CT is derived from an objective measure of behavior (change in length). Firstly, we now present change in length first. Secondly, we’ve added a “Strong CT threshold” bar to the change in length graphs so readers can see how Strong CT was calculated, and to better conceptually link the two charts. Please see each behavior figure.We have added optogenetic behavioral data from larvae carrying CIII-specific ChETA in order to contextualize the Aurora-evoked behaviors. Please see Figure 5, and Figure 5—videos 1-3.We have added an assessment of locomotor ability in the ANO knockdown animals, to test the plausible alternative hypothesis that knockdown generally affects contractile behavior. This uses a more comprehensive automated tracking tool (FIMtrack). Please see Figure 4 C-G.

Reviewer #3 (Recommendations for the authors):A specific suggestion for testing the proposed role for ca^2+^ in the model in Figure 8 would be to put BAPTA-AM into the recording pipette and/or bath, which should reduce cold-induced activity if ca^2+^-activated Cl^-^ conductances are important for this response.

We have performed a new experiment based on the reviewer’s suggestion, where we recorded cold-evoked neural activity in the presence of BAPTA-AM. In summary, BAPTA-AM application caused decreased cold-evoked activity.

The mentions of mammalian neuropathic pain papers related to spinal cord injury (SCI) on p.18 and elsewhere are not germane because the cited studies examined spinal neurons rather than primary sensory (DRG) neurons, they failed to describe "spontaneous nociceptor activity," and they described results from peripheral neuropathic models rather than SCI models. More appropriate would be references to peripheral neuropathic pain models that examined DRG neurons (e.g., PMID 24813295 and PMC6990784).

We have now cited these studies in the text.

p.3, line 57 – There appears to be a typo. What does "variably selection" mean?

This was a typo. We’ve fixed it to “variably selective,” referring to the different ion selectivity of TRP channels, across the superfamily.

p.4, line 78 – Again, there is little evidence for or against upregulation of NKCC1 in sensory neurons contributing to human neuropathic pain after SCI.

We have modified this in the spirit of the reviewer’s other suggestions.

p.10, Figure 2 headings – The data presented show that anoctamins contribute to the effects shown, but not that they are "required" for either CT responses or electrical activation by cold.

We have modified the Figure 2 headings. The new Figure 2 result headings are now “subdued and CG15270 (white walker) mutants respond less strongly to cold” (moved from Figure 1), and “subdued and white walker function in CIII cold nociceptors.”

p.12, Figure 3A – The y-axis label reads "trails" rather than "trials".

We have fixed this typo.

p.15, line 256 – For clarity the sentence should add "compared to the average expression levels for the whole body," or something equivalent.

We have added equivalent text.

p.15, line 274 – The first sentence should read "… selectively required for responses to cold stimulation." The next sentence would be clearer if it read "… alternative hypothesis that Cl^-^ currents in CIII neurons are inhibitory but have excitatory downstream effects; for example, …"

We have modified the text at the reviewer’s suggestions.

p.16, line 286 – The alternative hypothesis is not disproved, but the Aurora findings do provide strong evidence against that hypothesis.

We have revised this entire paragraph, and now omit the word(s) at issue.

p.18, lines 310+ … – See weakness #5 above.

We hope we are correct that the reviewer is referencing weakness #4. Throughout the revised text we moved away from terms like “hypersensitization,” and instead use “sensitization” and “hyperexcitability,” depending on if the context is behavioral or neural, respectively. This is all, of course, in addition to the specific edits made above and below.

p.20, Figure 7 – The phrase "causes CIII neuropathic sensitization" in the heading is incorrect because there is no reason to assume that overexpression of ncc69 causes neuropathy. What is shown is neuronal sensitization. A better title would be "Overexpression of ncc69 sensitizes and induces ongoing activity in CIII neurons."

We have modified the Figure 7 headings as follows: “Overexpression of *ncc69* increases CIII activity,” and “Overexpression of *ncc69* sensitizes larvae to less-noxious cold.” We’ve also modified the title of the manuscript, replacing “neuropathic” with “nociceptive.”

p.24, line 419 – References should be cited for this claim.

We have clarified and referenced Mòdol et al., 2014.